



# Altimetric observation of wave attenuation through the Antarctic marginal ice zone using ICESat-2

Jill Brouwer[1,2], Alexander D. Fraser[2], Damian J. Murphy[3,2], Pat Wongpan[2], Alberto Alberello[4], Alison Kohout[5], Chris Horvat[6], Simon Wotherspoon[1], Robert A. Massom[3,2], Jessica Cartwright[7], and Guy D. Williams[1]

[1]Institute for Marine and Antarctic Studies, University of Tasmania, Hobart,Australia
[2]Australian Antarctic Program Partnership, Institute for Marine and Antarctic Studies, University of Tasmania, Hobart, Australia
[3]Australian Antarctic Division, Kingston, Australia
[4]University of East Anglia, Norwich, United Kingdom
[5]National Institute of Water and Atmospheric Research, Christchurch, New Zealand
[6]Brown University, Providence, USA
[7]Spire Global, Inc., Glasgow, United Kingdom

**Correspondence:** Alexander Fraser (Alexander.Fraser@utas.edu.au)

**Abstract.** The Antarctic marginal ice zone (MIZ) is a highly dynamic region where sea ice interacts with ocean surface waves generated in ice-free areas of the Southern Ocean. Improved large-scale (satellite-based) estimates of MIZ width and variability are crucial for understanding atmosphere-ice-ocean interactions and biological processes, and detection of change therein. Legacy methods for defining the MIZ width are typically based on sea ice concentration thresholds, and do not directly
relate to the fundamental physical processes driving MIZ variability. To address this, new techniques have been developed to determine MIZ width based on the detection of waves and calculation of significant wave height attenuation from variations in Ice, Cloud and land Elevation Satellite-2 (ICESat-2) surface heights. The poleward MIZ limit (boundary) is defined as the location where significant wave height attenuation equals the estimated satellite height error. Extensive automated and manual acceptance/rejection criteria are employed to ensure confidence in MIZ width estimates, due to significant cloud contamination
of ICESat-2 data or where wave attenuation was not observed. Analysis of 304 MIZ width estimates retrieved from four months of 2019 (February, May, September and December) revealed that sea ice concentration-derived MIZ width estimates were far narrower (by a factor of ∼7) than those from the new techniques presented here. These results suggest that indirect methods of MIZ estimation based on sea ice concentration are insufficient for representing physical processes that define the MIZ. Improved measurements of MIZ width based on wave attenuation will play an important role in increasing our understanding
of this complex sea ice zone.



## 1 Introduction

Understanding the nature and drivers of the Earth's sea ice system (and change and variability therein) is a high priority in climate science (Meredith et al., 2019). Sea-ice-related processes play a crucially important role in Earth's climate system by

modifying and modulating interactions of the ocean and atmosphere, and by influencing the oceanic uptake and storage of anthopogenic heat and $CO_2$ from the atmosphere (e.g., Butterworth and Miller, 2016). Moreover, sea ice forms a key habitat for a diverse range of marine biota, from micro-organisms to whales (Massom and Stammerjohn, 2010).

An important element of this complex air-sea ice-ocean interaction system is the outer part of the sea ice zone, termed the marginal ice zone (MIZ). The MIZ is defined as the area where sea ice properties are impacted by open ocean processes,

especially ocean surface gravity waves (Wadhams, 1986). Wave-ice interactions are mutual: waves alter sea ice properties through physical and thermodynamic processes, and energy transferred while doing so attenuates wave amplitude by scattering and dissipative processes. Sea ice acts as a low-pass filter, preferentially attenuating higher frequency waves at a rate dependent on the sea ice physical properties (e.g., concentration, thickness, floe size; Squire, 2020). In some cases, long-period surface gravity waves have been observed to penetrate hundreds of kilometers into sea ice before their energy is fully attenuated (Liu

and Mollo-Christensen, 1988; Asplin et al., 2012; Stopa et al., 2018a), where they substantially impact the sea ice cover and the size distribution of ice floes. This is especially the case in the circum-Antarctic sea-ice zone, where long period and high amplitude waves from the surrounding high-energy Southern Ocean (Young et al., 2020) attenuate within the MIZ (Weeks, 2010; Horvat et al., 2020; Alberello et al., 2021).

The highly dynamic nature and intense ice-atmosphere-ocean interactions occurring in the MIZ have important effects on

sea ice properties and distribution, the structure and properties of the ocean and atmosphere, weather patterns, regional and global climate, and important marine ecosystems (Massom and Stammerjohn, 2010). Sea ice formation and melt processes within the MIZ are also a major driver of distinct regional patterns observed in Antarctic seasonal sea-ice advance and retreat (Kohout et al., 2014), and observed change and variability therein (Lubin and Massom, 2006). A primary process for wave alteration of sea ice coverage, properties, dynamics and thermodynamics is through wave-induced break-up caused by flexural

strain (Wadhams et al., 1986; Dumont et al., 2011; Bennetts et al., 2017; Wadhams et al., 2018). Wave-induced break-up and subsequent wave attenuation result in smaller floes close to the ice edge, due to breakup at the margins by larger amplitude waves, and progressive attenuation of wave energy (and larger floes) deeper into the ice pack (Collins et al., 2015; Fox and Haskell, 2001; Massom et al., 1999; Toyota et al., 2011). Floe size distribution is an important determinant of lateral sea ice melt, as the increased total perimeter of smaller floes enhances melt (Maykut and Perovich, 1987; Steele, 1992) and can

energize ocean eddy variability, driving faster sea ice retreat (Horvat et al., 2016). This process makes a major contribution to the rapid annual retreat of Antarctic sea ice each spring-summer. The presence of waves also determines ice type, prohibits floe welding – a process whereby small floes grow laterally and join into larger ones (Lange et al., 1989), whilst wave over-wash may also enhance melt (Massom and Stammerjohn, 2010). These processes present a positive feedback whereby less extensive sea ice coverage reduces wave attenuation, enhancing floe break-up and melt, thus underscoring their importance (Roach et al.,

2018; Alberello et al., 2019).





The MIZ, as viewed from remotely sensed sources, is historically defined using satellite passive microwave sea ice concentration (SIC) data as the area between the ice edge (15% SIC) and close ice as defined by the World Meterological Organisation (2014) (80% SIC; Strong, 2012; Strong et al., 2017). This definition allows large-scale delineation of this zone on daily timescales. However, mapping and monitoring the MIZ based on intermediate values of SIC is not physically-based, as it does not directly represent wave-ice or other coupled interactions. Indeed, SIC is influenced by a wide range of concomitant processes including winds and ocean currents, air temperature, upper ocean heat storage, turbulent and radiative heat exchange (Wadhams, 1986) and snow cover (Sturm and Massom, 2017). SIC-based MIZ retrieval is thought to incorrectly represent the true MIZ extent, particularly in the Antarctic (Vichi, 2021), as visible wave penetration can occur in areas of 100% ice coverage (Liu and Mollo-Christensen, 1988; Vichi et al., 2019), and, conversely, waves may not be present in low-concentration sea ice.

Altimetry holds the potential to measure wave propagation and attenuation in sea ice. Attenuation of waves in ice has been measured using satellite radar altimetry as early as the mid-1980s (Rapley, 1984). The ground resolution achievable by present radar altimeter technology is not sufficient to directly resolve wave attenuation through the MIZ. This is expected to be resolved with the proposed Surface Water and Ocean Topography (SWOT) radar interferometry mission (Fu and Ubelmann, 2013) due to be launched in late 2022 (Armitage and Kwok, 2021). Laser altimeters such as the Advanced Topographic Laser Altimeter System (ATLAS) instrument on board ICESat-2 (IS-2 hereafter) can observe at a sufficient resolution to resolve wave propagation (Horvat et al., 2020).

The combination of high vertical precision and horizontal resolution enables IS-2 to detect waves as they propagate into and flex sea ice (Horvat et al., 2020). With satellite ground speeds in excess of 7 km/s, this further enables near-instantaneous snapshots of wave attenuation with distance into the sea ice. For example, a 2019 storm in the Barents Sea was observed to generate waves in sea ice with heights above 2 m, which decayed over distances of several hundred kilometers into the sea ice near Svalbard (Horvat et al., 2020). With IS-2 operational since October 2018, and orbiting the Earth 15 times per day, the IS-2 dataset provides global coverage and combined with information about along-track floe sizes, concentrations, and thicknesses can provide a unique capacity for capturing information about wave attenuation for climate models (Tilling et al., 2018; Horvat et al., 2019; Roach et al., 2019; Horvat and Roach, 2021).

The variety of interactions between ocean surface waves and sea ice provides a strong motivation to observe, understand, simulate and predict current and future MIZ conditions and processes. Southern Ocean wave height is predicted to increase over the next 250 years (Dobrynin et al., 2012) as the frequency and intensity of storms increase, which will allow waves to ingress further into the MIZ (e.g. Squire, 2020), potentially increasing MIZ areal coverage, properties and influence. Being able to understand large-scale MIZ dynamics is an essential step to improving our understanding of the likely response of Antarctic sea ice to climate change, and its wide-ranging impacts. The purpose of this study was to develop a new method of determining the width of the outer Antarctic MIZ, by directly detecting the presence of waves (and their attenuation) in IS-2 surface height data. Large-scale (remotely sensed) MIZ width estimates based on direct detection of waves are expected to provide an improved spatial representation of the extent of this zone and the physical dynamics occurring within.

A recent analysis of IS-2 data measured the presence of ocean waves in ice by determining the presence of negative heights (after a mean sea surface correction was applied; Horvat et al., 2020). The results of this preliminary analysis reported that the



wave-affected MIZ extents were smaller than that defined by SIC. This is contrary to suggestions that SIC may underestimate MIZ extent due to the observed presence of surface gravity waves where SIC is 100% and therefore not classified as MIZ based on the SIC definition (Vichi et al., 2019). The Horvat et al. (2020) study required wave heights to be large relative to background sea ice and ocean variability, highlighting the need for spectral analysis of IS-2 heights to facilitate the separation

of wave presence from sea ice variability. To address this limitation, this study will aim to:

1. improve estimates of wave presence and attenuation in the Antarctic MIZ using spectral and spatial domain analysis techniques;

2. validate IS-2-derived significant wave height against wave buoy measurements; and

3. calculate MIZ width and compare to that derived from SIC to address the proposed hypothesis that the SIC-based

technique underestimates MIZ width compared to a wave attenuation-based definition.

## 2    Datasets

IS-2 was launched in September 2018 and provides coverage of the Antarctic MIZ along predominantly north/south track lines. IS-2 orbits at an altitude of ∼480 km, with 17 m diameter laser footprints spaced ∼0.7 m along track, arranged in a six-beam configuration. The standard deviation in vertical photon height measurements is of the order of centimeters (Neumann

et al., 2019). Higher-order sea ice height products are derived by accumulating 150 photon returns into approximately 10-20 m segments, with a reported along-track vertical precision of approximately 2 cm for Arctic sea ice (Kwok, 2019). Wave presence in the MIZ was determined from the variability of IS-2 reported surface heights. The IS-2 dataset used here was the Level 3 sea ice height product (ATL07, version 2), from the National Snow and Ice Data Centre (NSIDC; https://nsidc.org/data/atl07). The ATL07 algorithm corrects surface heights for deviations due to solid Earth tides, solid Earth pole tides, local displacement

due to ocean loading, atmospheric delay and mean sea surface (predetermined from IS-2 and Cryosat-2 data), ocean tides, long period equilibrium tides and geoid undulations (Kwok, 2019). ATL07 surfaces are produced where passive-microwave-derived sea ice concentration is equal to or greater than 15%.

As a proof of concept, we consider here all IS-2 tracks within four study periods of February, May, September and December of 2019. These were chosen to represent times of minimum extent, rapid advance, maximum extent and rapid retreat,

respectively.

SIC-based estimates of MIZ distance were also computed for comparison with along-track spectral information. We use the ARTIST Sea Ice algorithm daily 6.25 km SIC data (Spreen et al., 2008) downloaded from https://seaice.uni-bremen.de/sea-ice-concentration/amsre-amsr2/, rather than the NSIDC ice concentration product packaged with IS-2 data, due to its higher resolution facilitating finer-scale consideration of wave attenuation.

To validate IS-2-retrieved wave information, we compare IS-2-derived significant wave height ($H_s$) estimates to measurements made by five wave-ice interaction buoys. The buoys, which were manufactured by P.A.S. Consultants P/L, use the Sparton AHRS-M1 micro-sized, light weight, low power inertial sensor with a built-in adaptive calibration mode. The buoys




were designed for sea ice deployment, and monitor acceleration in all planes. Data bursts (acquisitions at a rate of 64 Hz) were separated by 640 seconds. A low-pass, second-order Butterworth filter was applied with a cut-off at 0.5 Hz, and subsampled

to 2 Hz. A high-pass filter was then applied and the acceleration integrated twice to provide the displacement. Calculation of spectral density was performed using Welch's method (Welch, 1967), with a 10% cosine window and de-trending on four segments (each 256 s long) with 50% overlap. Spectral moments were also calculated, and $H_s$ was obtained from the zeroth spectral moment, defining the total variance (or energy) of the wave system within the frequency range detectable by the buoy. Five buoys were deployed on 2019-12-09 and 2019-12-10, from north (64.27° S) to south (64.75° S) along ~120.5° E. Three

of these buoys were deployed near the ice edge, one in low sea ice concentration and another in high sea ice concentration. For all deployments, the sea ice primarily consisted of pancakes with gaps generally filled with frazil or brash ice. In total, 4,402 wave records were captured over six months (from 2019-12-10 to 2020-06-12).

## 3  Methods

For each track, preliminary quality control of the IS-2 heights was first undertaken. Segment heights >100 m were removed.

Each track line was split into descending and ascending orbit components, and ascending tracks were reversed so that all analyses were undertaken from north to south. As a consequence, waves generated from the limited fetch within coastal polynyas were ignored here.

Surface height data were interpolated onto a regular 8 m grid format using a cubic spline method in order to provide equally-spaced points for application of amplitude scaling corrections due to cloud-obscured data (detailed in Appendix A).

Non-Uniform Fourier Transform (NUFT) techniques including the Lomb-Scargle periodogram, or the method of Greengard and Lee (2004) suggested by Horvat et al. (2020) are not considered here.

Interpolated along-track heights were divided into windows of 6.25 km, to provide a similar resolution to the ASI/ASMR-2 SIC product for comparison. Sections were selected for spectral analysis in 6.25 km sliding windows with a 1 km step (a window overlap was implemented). The maximum allowable amount of missing data (due to cloud contamination) in each

window was set to 50% (Murphy et al., 2007).

SIC-based MIZ width is defined as the distance between the 15% and 80% SIC contours (Strong et al., 2017). Here we calculate MIZ width from SIC along the IS-2 tracks rather than using, for example, meridional transects (Stroeve et al., 2016) or more sophisticated mathematical techniques (see recommendations in Strong et al., 2017) to facilitate direct comparison between SIC- and IS-2-derived estimates. Secondary occurrences of lower SIC (<80%) further south than the northernmost

80% boundary were not included in the SIC MIZ width calculations, to remain consistent with the fact that the inner MIZ was not measured by the wave attenuation methods. "Effective penetration distance" ($x_e$) was calculated by integrating the SIC



$(P_z)$ from the ice edge (0) inwards to point $x$ (after Wadhams, 1975), and is referred to henceforth as "corrected distance into the MIZ":

$$x_e = \int_0^x P_z dz. \tag{1}$$

This corrected distance metric represents the equivalent distance a wave would have to penetrate if the ice from the edge to distance $x$ were consolidated to 100% SIC, i.e., is always shorter than the physical distance.

## 3.1 Spectral and spatial domain analyses

Spectral analysis was completed for each suitable section and along each track. The effect of lost variance due to windowing and missing data (i.e., cloud cover) was corrected using $W_{ss}$ scaling described in Appendix A. Two windowing functions,

boxcar and Hann (Earle, 1996) were tested to determine the effects of spectral leakage on spectral amplitude estimates. Each window was combined with the missing data profile for each section and the $W_{ss}$ scaling factor calculated from this combined window.

Sampling effects of non-random data gaps may contribute their own spectral characteristics in addition to those from the surface height data itself (Murphy et al., 2007). To circumvent this, a Spatial Domain Filtering (SDF) spectral analysis method

using Finite Impulse Response Filters (FIRFs) was also employed, following Murphy et al. (2007). An additional advantage of applying filtering in the spatial domain is the ease with which filtered data can be inspected. Here the FIRFs are a set of Gaussian functions (in the spectral domain), with a constant Q factor (Palo et al., 1998) (here Q=2.25, encompassing three complete wave cycles). A bank of 11 filters with center wavelengths ranging from 38 m to 1500 m was originally considered (i.e. wave periods from 5 to 31 s), in accordance with expected wavelength values of surface gravity waves (Toffoli and Bitner-

Gregersen, 2017). Inter-filter spacing was equivalent to the filter bandwidth. The filtering process involved the convolution of the height data with the spatial domain filter function generated from the FIRFs.

A subset of four contiguous filters (with peak wavelengths of 165, 239, 345 and 498 m, roughly equivalent to 10, 12, 15 and 18 s periods) was subsequently chosen from this filter bank for final MIZ width retrieval, based on the results of Stopa et al. (2018b). Filters with center wavelengths shorter than 150 m were not considered representative of MIZ width due to rapid

attenuation of shorter wavelengths. These wavelengths may also be associated with roughness due to ice features, potentially confounding $H_s$ estimates in the inner MIZ. Wavelengths longer than $\sim$500 m were not considered in MIZ width estimation here, as they have a weaker physical effect on sea ice (e.g., a lower modelled break-up stress; Montiel and Squire, 2017).

## 3.2 Derivation of significant wave height

Significant wave height was used as a metric to measure wave attenuation in the MIZ, and is related to wave energy (Kohout

et al., 2020). Four different measurements of significant wave height were calculated for each suitable along-track section: 1) Hann- and 2) boxcar-windows moment-based; and standard-deviation based estimates from 3) interpolated and 4) SDF-filtered





height series. Hann- and boxcar-windowed power spectra were bandpass-filtered from the Nyquist wavenumber to 1,500 m to remove longer wavelength signals (e.g. tides, geoid variations) that may have remained despite IS-2 corrections (Kwok, 2019). Significant wave height was then calculated from the zeroth moment ($m_0$) of each power spectrum after bandpass filtering.

Significant wave heights calculated in this way were termed ($H_{m0}$):

$$H_{m0} = 4\sqrt{m_0}. \tag{2}$$

Spatial domain estimates of significant wave height ($H_s$) were also determined from the standard deviation ($std$) of the interpolated and SDF-filtered along-track height data ($h$)

$$H_s = 4 \cdot std(h). \tag{3}$$

Mean significant wave height for each 1 km along-track spacing was calculated by averaging across the three corresponding $H_s$ estimates from each beam. Error in mean significant wave height was calculated by quadrature addition of the standard deviation of the three $H_s$ values, and the mean IS-2 height error within each 6.25 km section.

### 3.3 Attenuation curve fitting and MIZ width estimation

The MIZ width was estimated by fitting segmented linear regressions to $H_s$ and $H_{m0}$ transects with the R package "segmented"
(Muggeo, 2003) to automatically divide transects into outer "attenuation-dominated" and inner "ice structure-dominated" sections. Reliable initial estimates of the breakpoint between these two sections were obtained by fitting thin-plate regression splines (Wood, 2003) using the R package "mgcv" (Wood, 2017), and estimating the breakpoint as the first local minimum of the fitted spline.

Following automated definition of the breakpoint between these regions, $H_{m0}$ and $H_s$ attenuation within the outer region
were quantified in order to determine MIZ width. Attenuation of significant wave height in sea ice has been reported as exponential as well as linear (Kohout et al., 2014). Here, we model attenuation using both models (noting that analysis of the falloff coefficient or exponent is outside of the scope of the present work). The inner boundary of the MIZ was defined as the point where the modelled significant wave height intercepted the quadrature-added error of the three $H_s$ estimates (one per strong beam) and the estimated error in segment height. Concentration-corrected distance from the ice edge was used for MIZ
width determination (and this was later converted back to physical distance for reporting MIZ width). In these measurements of attenuation, an along-track wave propagation direction and stationarity were assumed. Wave propagation direction assumption caveats are given in the Discussion section.

### 3.4 Track selection during processing

Not all IS-2 tracks were able to be processed in this way. Appropriate track selection criteria consisted of two components:
firstly, the (automated) identification of tracks that contained enough cloud-free data to identify the presence or absence of

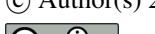



wave attenuation throughout the MIZ (Figure 1a, c and f); and secondly, assessment of whether or not a track contained characteristics required to be identified as MIZ or not (both automated and manual components; Figure 1h and i, respectively). Any thresholds were chosen so as to be conservative (i.e., to ensure confidence in $H_s$ and MIZ width estimates by discarding tracks without apparent $H_s$ attenuation). These procedures are described in detail below, and caveats associated with manual

track selection given in the Discussion section.

Any tracks with excessive cloud coverage were automatically identified and rejected from further processing. In this first step, IS-2 tracks were either accepted or rejected based the following criterion: tracks with ≥50% data present in the central strong beam within either 100 km or 500 km from the ice edge were accepted for further processing (Figure 1a). Tracks failing this criterion were rejected. The 100 km bound was chosen to allow selection of records of the outer MIZ and allow MIZ width

estimation at times of reduced sea ice extent (and hence MIZ) around the sea-ice minimum (February-March), while the 500 km bound was chosen so as to not exclude lines where a deeper MIZ may be present.

Further filtering steps in the first stages of processing (Figure 1a-d) were undertaken as follows:

1. Tracks acquired during satellite reorientation were excluded since data quality may be degraded (Figure 1a).

2. Tracks were excluded if all sections violated the maximum missing data threshold for spectral analysis (Figure 1c).

3. All tracks were excluded in the region from 50 to 61° W – a region of persistent multi-year sea ice near the ice edge to the east of the Antarctic Peninsula. Considerable roughness of the multi-year sea ice in this area had the potential to confound the automated partitioning between "attenuation-dominated" and "ice structure-dominated" regions outlined above.

Accepted tracks were required to contain at least ten valid 6.25 km sections for automated break-point segmentation (Figure

1f). Following this step, any tracks with a positive gradient in the outer "attenuation-dominated" region were discarded. If the estimated MIZ width was larger than the sea ice zone width (i.e., due to $H_s$ attenuation extrapolation in regions of data gaps) the track was discarded (Figure 1h). As a result, cases of complete MIZ from the ice edge to the continent are likely to be erroneously discarded (a condition likely to occur in the narrow sea ice zone throughout much of East Antarctica or in the Bellingshausen Sea (Massom et al., 2008). Method improvements are required to account for these cases, and will be discussed

later.

In order to ensure accurate MIZ width estimation, manual selection was undertaken to remove cases where no wave attenuation was apparent (largely arising due to cloud cover over the region experiencing attenuation) or the attenuation models were clearly incorrectly fit (primarily due to $H_s$ contributions from ice structure in the "attenuation-dominated" region; Figure 1i). Wave attenuation was manually assessed by identifying the presence or absence of a triangular shaped "envelope" of wave

decay in the height data, showing as large positive and negative heights at the ice edge which attenuate with increasing distance into the sea ice (Horvat et al., 2020). The next criterion for this manual assessment was that the break-point of the piecewise regression occurred at the transition from "attenuation-dominated" to "ice structure-dominated" regions. To avoid high uncertainty in MIZ width estimation due to missing (cloud-masked) data, tracks were further excluded if the boundary between



**Figure 1.** Flow chart of track selection criteria and processing steps.

**Track rejection/acceptance criteria**

**IS-2 processing steps**

**a) Preliminary acceptance**
>50% data within 100 km OR 500 km from ice edge, satellite orientation not transitioning

b) Interpolation, section selection, distance from ice edge calculated

c) At least 1 6.25km section with SIC>15% and >50% data

d) Spectral and FIRF *Hs* values estimated for cloud-free sections

f) Local minima present, at least 10 data points either side of local minima

e) GAM smoothing of mean *Hs*, position of first local minima estimated

g) Weighted piecewise regression fit to *Hs* and log(*Hs*) where distance <= 2x local minima

**h) Automated acceptance:**
- Breakpoint present
- Negative slope for northern regression line
- Predicted MIZ distance within distance range of IS-2 data

**i) Manual acceptance:**
- Negative *Hs* visible in interpolated height data
- Piecewise regression fit correctly to attenuation and ice structure components
- Inner MIZ apparent

j) MIZ width estimated, regression parameters and associated errors output





the "attenuation-dominated" and "ice structure-dominated" regions was obscured (by cloud). Due to the considerable manual
overhead, four months of 2019 were analysed and presented here. These months (February, May, September and December
of 2019) were selected to represent phenologically-important periods of sea ice minimum extent, rapid growth, maximum extent and rapid retreat, respectively (Eayrs et al., 2019). Table 1 gives the number of tracks prior to track selection, and after
automated/manual selection in each month.

| Selection step | February | May | September | December | Total |
|---|---|---|---|---|---|
| Prior to selection | 784 | 868 | 840 | 868 | 3360 |
| Automated selection | 219 | 408 | 342 | 252 | 1221 |
| Manual selection | 24 | 167 | 101 | 27 | 320 |

**Table 1.** Number of tracks in each month (of 2019) remaining after automated and manual filtering was applied. Tracks are split into
ascending and descending components, i.e., there are two tracks per data file.

For the purposes of validation of IS-2-retrieved attenuation, $H_s$ was directly measured from a deployment of five wave-sensing buoys, and used to validate the $H_s$ estimates derived from IS-2. Co-locations of IS-2 tracks and wave buoys were
first identified, prioritising temporal proximity (within 6 hours) over spatial proximity (within 400 km) under the assumption
that wave conditions de-correlate quickly with time (a lag of 6 hours reduces the autocorrelation coefficient to between 0.79
and 0.95 for the five buoys in this buoy deployment). Tracks containing a buoy co-location were then analysed to find the
closest (spatial) measurement of $H_s$ for comparison. IS-2-derived $H_s$ measurements at these locations were also compared
to a modified version of the Horvat et al. (2020) technique for estimating wave-affected fraction. The modification allows
along-track (rather than gridded, as published) estimates of wave-affected fraction using each IS-2 beam, along a 50 km sliding
window.

## 4 Results

Two case studies (Table 2) are presented to demonstrate the methods involved in directly detecting the presence of waves in
the IS-2 height data, and measuring their attenuation to find the inner MIZ boundary. Cases from September and February in
2019 are chosen.

| Date | ATL07 track ID | Latitude | Longitude |
|---|---|---|---|
| 2019-09-09 | 20190909191519_11260401 | 65.0° S | 132.2° W |
| 2019-02-04 | 20190204191931_05860201 | 68.8° S | 21.1° E |

**Table 2.** Summary of the two case study tracks. Latitude and Longitude of the along-track ice edge location are provided.





**Figure 2.** Panel a: Resampled segment heights from the central strong beam, for case study 1. Panels b-e: SDF-filtered heights for each of the four FIRF filters used in MIZ width estimation, for the first 200 km of the track.

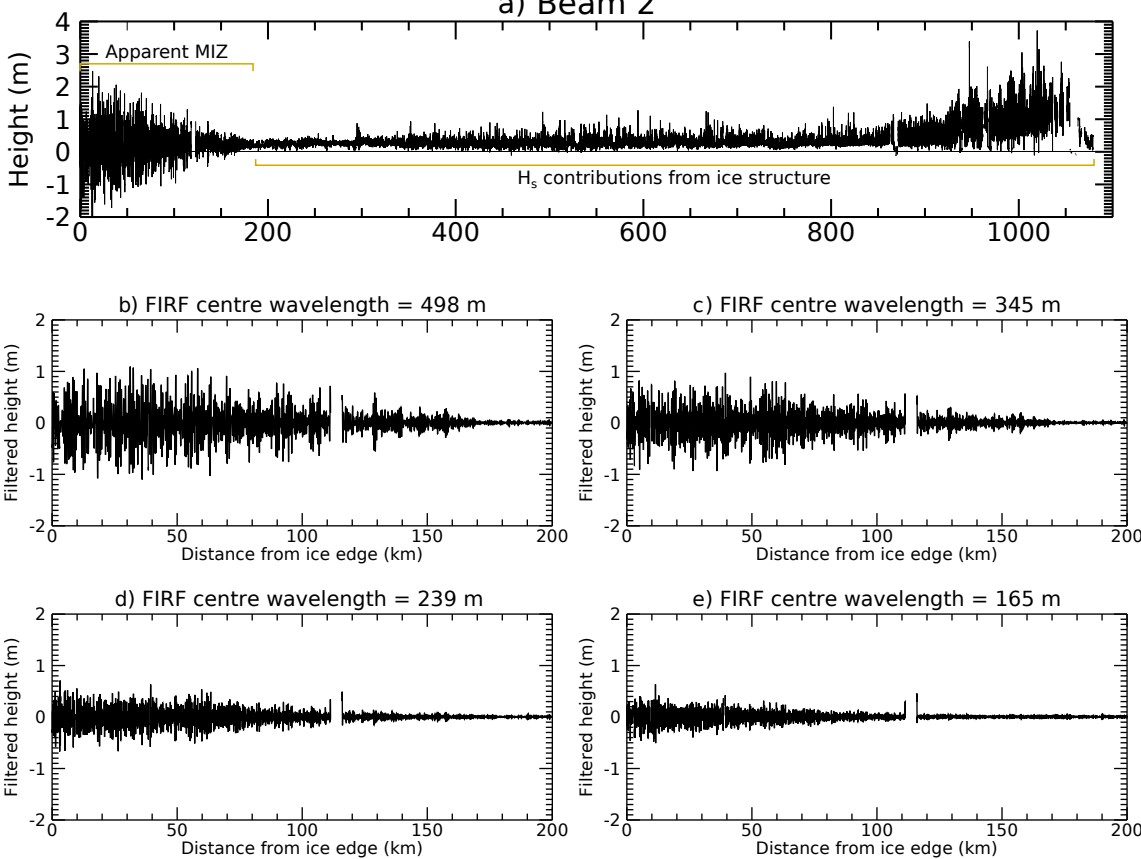

## 4.1 Case study from 2019-09-09

Figure 2a shows IS-2 heights observed from the ice edge to the continent. Negative heights indicative of wave passage (Horvat et al., 2020) were visible in all three strong beams (only centre beam shown) within ∼180 km of the ice edge. This region
exhibits a distinctive attenuation envelope. Note that most height contributions due to tides, inverse barometer effects, geoid undulations and mean sea surface have been removed by the ATL07 algorithm (Kwok, 2019). Although some residual remains, these corrections are sufficient for the purposes of visual detection of wave attenuation during the manual filtering in this study. Increasing heights at around >800 km from the ice edge indicates a transition to thicker ice.

SDF-filtered altimetric heights displayed varying signal amplitude (outer 200 km shown in Fig. 2b-e). In this case, the largest
heights were present in the data filtered with the SDF with a centre wavelength of 498 m. Waves of this wavelength appeared to penetrate up to ∼170 km from the ice edge.



**Figure 3.** Mean power spectral density as a function of wavenumber and distance from the ice edge, for case study 1.

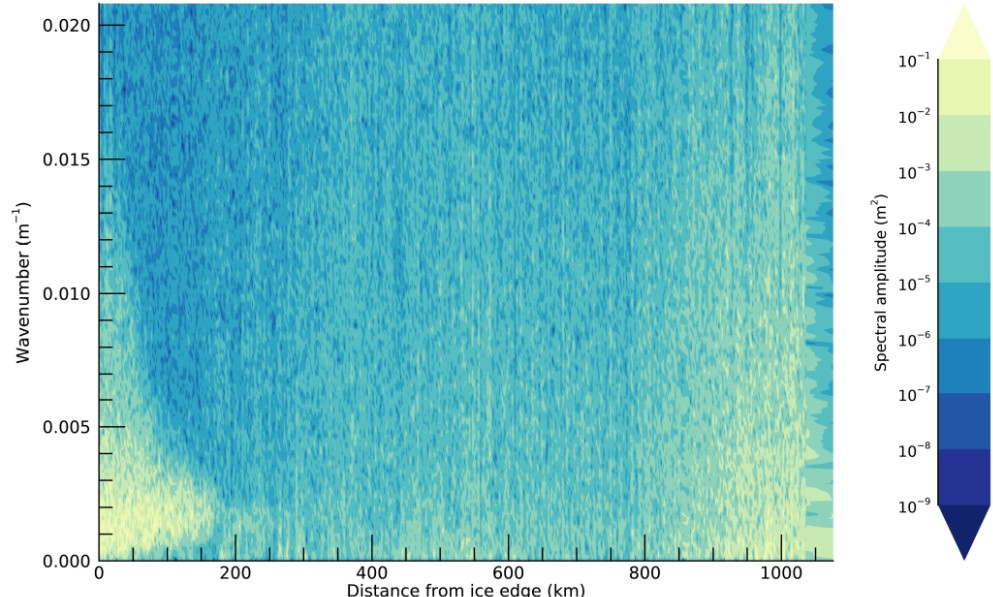

Figure 3 presents the power spectrum for the entire track. Narrowing of the spectrum from 0 to 200 km is evident. From ~250 to 800 km there is a region of low spectral amplitude due to low variability in height. After 800 km, high spectral amplitude is spread across a wider wavenumber ($2\pi$/wavelength) range. This distribution of spectral amplitude is very different to that of the first 200 km.

Power spectra of individual 6.25 km sections within 200 km of the ice edge (Fig. 4: Hann-filtered spectra shown only; boxcar-filtered spectra were similar) displayed a prominent peak wavenumber of ~0.0015 m$^{-1}$, corresponding to a wavelength of 650 m (wave period of ~20 s). This is similar to the dominant wavelength suggested by the FIRF analysis. With increasing distance from the ice edge, the amplitude and width of the spectral peak decreased, and the peak wavelength shifted towards longer wavelengths (smaller wavenumbers). These spectral changes match with those expected for wave attenuation in ice, where there is preferential attenuation of higher wavenumbers (Wadhams et al., 1988). Power spectra of sections close to the Antarctic continent (>817 km from the ice edge; given in Fig. 4b) peaked at very short wavenumbers, corresponding to very long wavelengths (Fig. 4b). In this region, the spectral shape displayed a near-monotonic decrease in spectral amplitude, characteristic of Brownian noise (Gilman et al., 1963). For this section of ice >817 km from the ice edge, total spectral amplitude increased with increasing distance from the ice edge as the variance in surface height increased.

The steps involved in attenuation model-fitting and MIZ width estimation are demonstrated in Figs. 5, 6 and 7 for all methods of MIZ width estimation presented here. The automated segmentation technique is able to effectively estimate the first local minimum (red dashed lines, Figure 5), corresponding to the transition from wave attenuation-dominated $H_s$ to ice structure-dominated $H_s$. $H_s$ was expected to approach zero when waves were fully attenuated (i.e. no surface variations in the absence





**Figure 4.** Hann-windowed power spectra of sections close to the ice edge (a) and in the inner pack (b), for case study 1. The colour scale indicates distance to the continent. Power spectral estimates were smoothed with a 5 point moving average. The black line in the upper right of panel b represents the spectrum of a Brownian noise signal (-20 dB/decade).

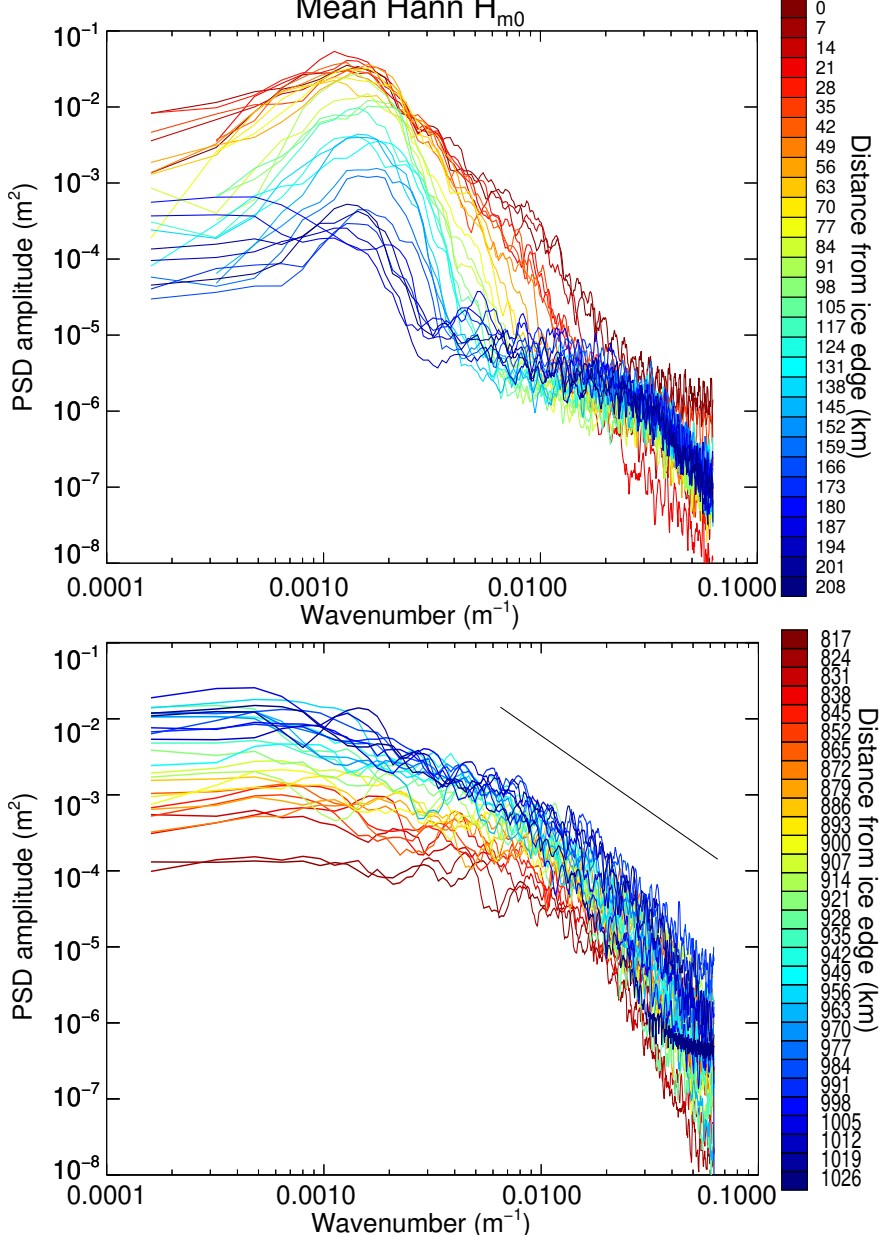





of waves). However for all $H_s$ estimation methods there was a positive offset in $H_s$ after the transition to the ice structure-dominated region, likely associated with $H_s$ contributions from variations in ice thickness. This offset was slightly lower for the SDF $H_s$ estimates than the other methods which measured the amplitude of the whole spectrum (Figs. 6 and 7). Model fit to the attenuation appears reasonable (Figures 6 and 7), and the linear attenuation model appears to fit the data better than the exponential model in this case. Assuming linear attenuation, MIZ width estimates range from 109 to 184 km, as estimated by the 126 m and 498 m SDFs, respectively (with $H_s$-derived estimates falling within this range). Exponential-modeled MIZ width estimates range from 179 km to 263 km.

To summarise the results of this case study, the presence of waves was apparent in the negative heights in the spatial data, and their attenuation was shown by the decrease in both positive and negative height values with increasing distances from the ice edge (the characteristic triangular envelope). This pattern of attenuation was visible until ∼180 km from the ice edge. Wave attenuation was also evident in the spectral domain, shown by the decreasing amplitude and narrowing of the spectral peak with increasing distance from the ice edge. SDF-based $H_s$ estimates of MIZ width ranged from 109 to 184 km (assuming linear attenuation, and depending on the centre wavelength of the spatial domain filter). The three spectral $H_s$-based estimates were also within this range.

Underestimation of the MIZ width using the SIC-based technique for this case is shown in Figure 8. In this figure $H_{m0}$ attenuation fit to distance from the ice edge (rather than using corrected distance) is given for visual comparison (for reference, the MIZ width estimated using the corrected distance metric was 177 km). Agreement between the distance over which waves were visible, and the MIZ width distance derived from wave attenuation (i.e., panels a and c) provide high confidence in the methods of MIZ width retrieval. A further case study, from 2019-02-04, is presented in the following section, as evidence of this technique working well during the summertime sea ice minimum.

## 4.2 Case study from 2019-02-04

The summer case study exhibited a narrower MIZ than the September case study presented above, and smaller wave amplitude relative to inner-MIZ and pack ice structure variability (Figure 9). Similar patterns of attenuation as those in the first case study were also present here in both spatial and spectral domains, i.e., the presence of negative heights in spatial data and a downshift and narrowing of the spectral peak (Fig. 11). The peak wavelength indicated by both the SDFs and Hann power spectra was ∼200 m (corresponding to a wavenumber of ∼0.03 and a period of 11 s; Figs. 9, 10 and 11), shorter than that of case study 1 by ∼450 m.

Breakpoint estimation also performed well for this case study, despite fewer points being available for fitting (Figs. 12, 13 and 14). Similar to the September case study, $H_s$ values did not reach zero at the end of wave attenuation (indicating ice structure contribution to apparent $H_s$ at the transition point), and the "offset" here was again smaller for the SDF-based techniques (∼0.2 m for SDF filtering-based $H_s$, vs ∼0.6 m for Boxcar $H_{m0}$, Hann $H_{m0}$ and standard deviation-derived $H_s$). Most methods resulted in an IS-2-estimated MIZ width of 30 to 40 km, with the exponential attenuation model resulting in wider MIZ estimates. As with the previous case study, the SIC-based technique considerably underestimates MIZ width (with the 80% SIC contour encountered at a distance of only 7 km from the ice edge; Figure 15).





**Figure 5.** GAM-based local minimum estimation for the spectral (left column) and SDF-derived (right column) estimates of $H_s$, for case study 1. The section from the ice edge to two times the GAM-based minumum distance is used for breakpoint estimation.





**Figure 6.** MIZ width estimation for the spectral $H_s$ estimation methods, for case study 1. Linear fit is shown on the left, and exponential (by log-transforming the y-axis) on the right. The dotted black line represents break point estimation of the piecewise linear regression. The blue line represents the quadrature-added uncertainty in $H_s$ estimates. The estimated MIZ width is marked by the red vertical line, and labelled in terms of the equivalent uncorrected (actual) MIZ width.





**Figure 7.** As for Fig. 6 but for SDF methods.



**Figure 8.** Direct comparison of along-track a) IS-2 beam 2 heights, b) SIC-based MIZ width and c) $H_s$ and the resulting MIZ width estimated using a wave attenuation-based definition, for case study 1. The case illustrated here shows MIZ width estimated using a linear fit to the Hann-windowed $H_{m0}$ values. Here, for comparison purposes, we perform the attenuation analysis (panel c) without consideration of SIC (i.e., on uncorrected distance).



**Figure 9.** Panel a: Resampled segment heights from the central strong beam, for case study 2. Panels b-e: SDF-filtered heights for each of the four FIRF filters used in MIZ width estimation.

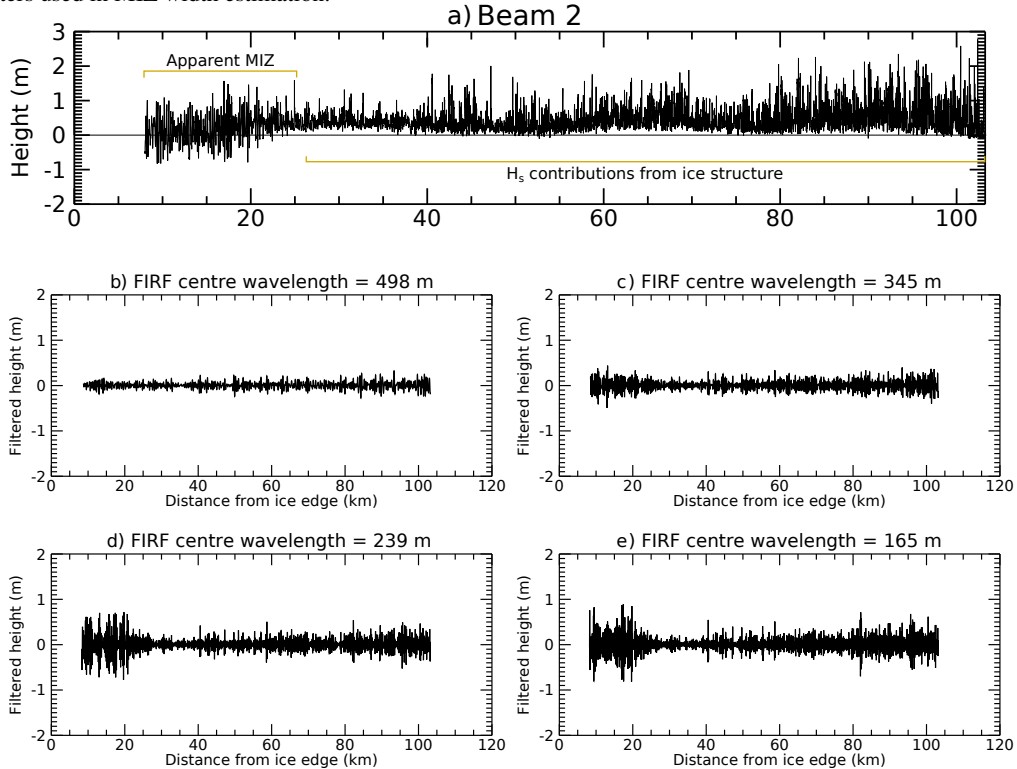

### 4.3 Validation of IS-2 $H_s$ against wave buoy-derived $H_s$

From 80 tracks with a spatio-temporal separation (between buoy- and IS-2-measured $H_s$) of less than 400 km and six hours, ten were selected for comparison, based on the IS-2 track selection criteria presented above, with most rejected due to cloud cover. The mean time separation between buoy data acquisition and the corresponding IS-2 track overpass was 38 minutes. Validation of IS-2-derived $H_s$ against buoy measurements is presented in Fig. 16, and was found to be very sensitive to their spatial separation (i.e., waves decorrelated quickly with distance). For tracks within 200 km of the buoy measurement, the

Pearson's correlation coefficient was r=0.94 (but with only n=3), however this remained high at r=0.72 when buoy-satellite conjunctions within 300 km were considered (n=6). The correlation is significant for spatial separation less than 300 km. For conjunctions within 300 km, the $H_s$ regression slope was 0.44, indicating that IS-2 underestimates $H_s$ by a factor of ~2.25. This is not surprising given the fundamental difference in measurement technique, however the high correlation and positive slope, particularly at < 300 km separation, gives confidence in the approaches demonstrated here. For the same ten IS-2 tracks,

the correlation between wave-affected fraction (from the along-track-modified Horvat et al. (2020) technique) and IS-2-based





**Figure 10.** Mean power spectral density as a function of wavenumber and distance from the ice edge, for case study 2.

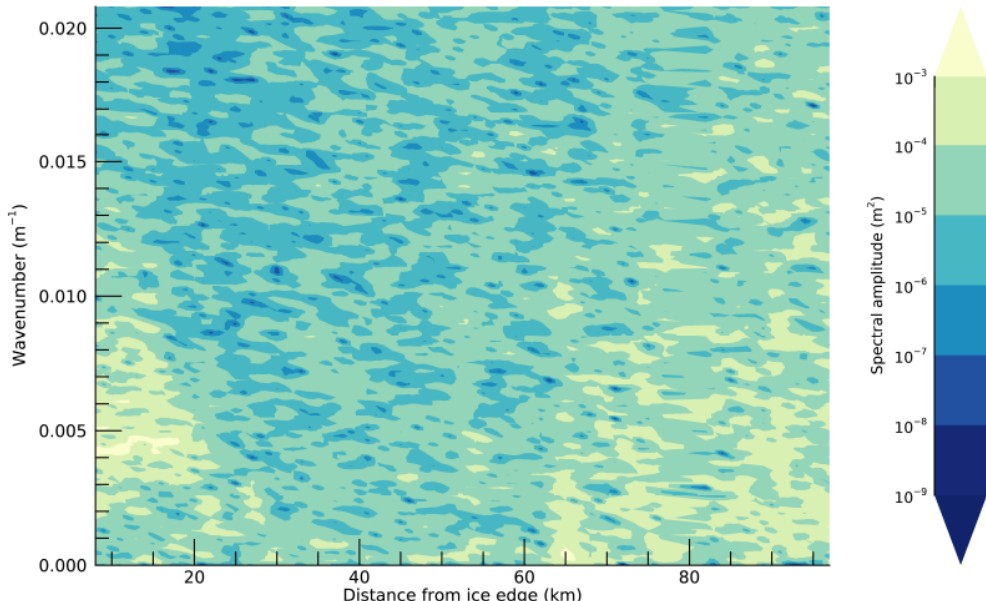

$H_s$ retrieved here was r=0.55 (p=0.049), indicating a significant correlation despite differences in wave estimation technique. More detailed comparison of these two IS-2-based wave estimation techniques is planned for the future.

### 4.4 Comparison of IS-2 MIZ width estimation techniques

Based on the case studies presented in 4.1 and 4.2, and other tracks assessed during manual filtering, the SDF technique was
chosen to estimate MIZ width. The median value of the MIZ distance determined in four SDF wavelengths (165, 239, 345 and 498 m) was used as the MIZ width metric to reduce the reliance on attenuation in one particular wavelength range. The advantages of the SDF technique over the spectral methods were two-fold: Selection of MIZ width estimates from SDFs in the wavelength range of 165 to 500 m enabled increased sensitivity to waves within a wavelength range that was assumed likely to physically impact the ice and prevalent in the Antarctic MIZ (see Figure 2 in Stopa et al., 2018a); and appropriate SDF
selection minimised $H_s$ contributions from ice structure. The simplicity of applying SDFs and relevant amplitude correction in addition to the absence of the effect of spectral noise caused by convolution of data gaps were further advantages (Murphy et al., 2007).

There was a very high correlation between SDF median-derived MIZ width estimates and those derived from all other $H_s$ estimation methods (Figure 17). Regression slope across all comparisons was in the range of 0.918 to 1.01, indicating robust
agreement between all MIZ width retrieval techniques. The Pearson's correlation coefficient between SDF-derived and all other techniques was between r=0.949 and 0.978.



**Figure 11.** Hann-windowed power spectra of sections close to the ice edge (a) and in the inner pack (b), for case study 2. The colour scale indicates distance to the continent. Power spectral estimates were smoothed with a 5 point moving average.





**Figure 12.** GAM-based local minimum estimation for the spectral (left column) and SDF-derived (right column) estimates of $H_s$, for case study 2. The section from the ice edge to two times the GAM-based minimum distance is used for breakpoint estimation.





**Figure 13.** MIZ width estimation for the spectral $H_s$ estimation methods, for case study 2. Linear fit is shown on the left, and exponential (by log-transforming the y-axis) on the right. The dotted black line represents break point estimation of the piecewise linear regression. The blue line represents the quadrature-added error of $H_s$ estimates. The estimated MIZ width is marked by the red vertical line, and labelled in terms of the equivalent uncorrected MIZ distance.

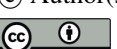

**Figure 14.** As for Fig. 13, but for SDF methods.



**Figure 15.** Direct comparison of a) IS-2 beam 2 heights, b) SIC-based MIZ width and c) MIZ width estimated using a wave attenuation-based definition, for case study 2. The case illustrated here shows MIZ width estimated using a linear fit to the Hann-windowed $H_{m0}$ values. Here, for comparison purposes, we perform the attenuation analysis (panel c) without consideration of SIC (i.e., on uncorrected distance).

### a) Beam 2 interpolated heights

### b) Along−track SIC

### c) Hann Hm0





**Figure 16.** Validation of IS-2-derived $H_s$ against wave buoy data, using Pearson's correlation coefficient (black line) and regression slope (red line), as a function of closest distance between track and buoy. The regression slope of less than unity indicates that IS-2 under-estimates $H_s$. The blue line indicates the correlation coefficient between the $H_s$ technique presented here and the wave-affected fraction using a modified along-track wave-affected fraction metric (after Horvat et al., 2020). The dashed line indicates the significance threshold for the blue and black lines.

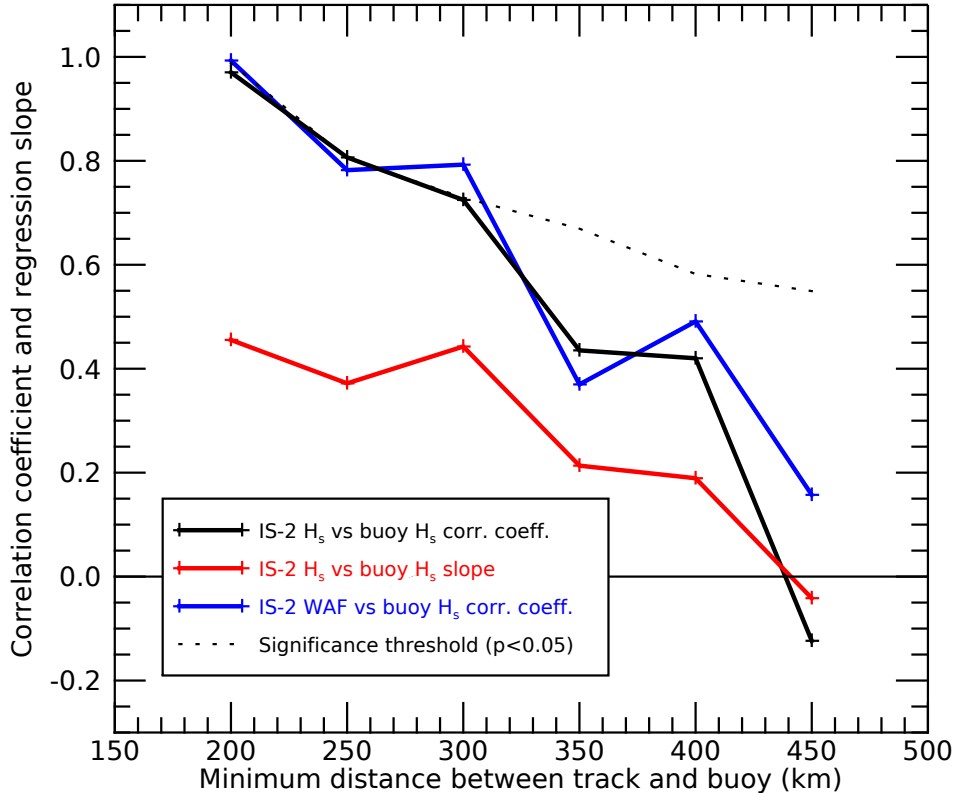

## 4.5 Attenuation model comparison and MIZ width uncertainty quantification

RMSE residual between observed $H_s$ and the fit in the attenuation-dominated region was approximately two times smaller for the linear attenuation model than the exponential fits over the four months in the study period (Table 3), indicating that the linear attenuation model provided a better fit for the cases studied here. MIZ width uncertainty, calculated from uncertainty in the y-intercept and slope of the fit (see Table 3), was larger for the exponential attenuation model in May and September, and ranged from ∼8% (linear; May and September) to ∼19% of the overall MIZ width (linear, February).

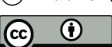



**Figure 17.** Linear regression between SDF-based MIZ width estimates and those based $H_{m0}$ with boxcar (top row) and Hann (middle row) windowing, and the $H_s$ estimate derived from the standard deviation of heights (bottom row). Spectral analysis methods were compared for both linear (left) exponential (right) attenuation modelling methods. Solid black lines show the linear regression (constrained to pass through the origin), with slope (m) and Pearson correlation coefficient (r) values indicated in the top left of each plot. Dashed lines represent a slope of unity.



| Month | RMSE (m) | | MIZ uncertainty (km, (%)) | |
|---|---|---|---|---|
| (2019) | Linear fit | Exponential fit | Linear fit | Exponential fit |
| February | 0.084 | 0.104 | 8.2 (18.9) | 6.6 (10.8) |
| May | 0.056 | 0.096 | 12.1 (7.9) | 34.7 (17.9) |
| September | 0.079 | 0.204 | 18.4 (8.8) | 32.5 (12.5) |
| December | 0.052 | 0.066 | 23.4 (16.0) | 13.1 (7.5) |
| Mean | 0.065 | 0.129 | 14.7 (9.0) | 29.9 (18.3) |

**Table 3.** Uncertainty statistics of SDF-median MIZ width estimates by month. MIZ uncertainty is the error in MIZ width calculated from the slope and intercept uncertainty (which were quadrature-added to the window size, 6.25 km), shown as absolute values (km) and as a percentage of the mean MIZ width.

### 4.6 SDF-based MIZ width estimation compared to the SIC-based technique

The traditional SIC-based MIZ width estimation gave lower MIZ width estimates than those derived from SDF $H_s$ attenuation
in all months analysed. A comparison of these techniques is shown in Figure 18 (exponential results have been omitted here
due to the larger attenuation model fit residuals and uncertainties described above, but are similar). SIC-derived and SDF-
derived MIZ width estimates were closest in February where linear SDF-derived MIZ estimates were ∼2.3 times larger than
SIC-derived estimates. SDF-derived estimates were ∼4.6 and 6.7 times wider in May and September, respectively. The largest
differences between estimates occurred in December, with a regression slope of 14.9.

### 4.7 MIZ width seasonality

For the same set of tracks (i.e., those which passed the automated and manual track selection mentioned above), median SIC-
derived MIZ width estimates were far narrower than SDF-derived MIZ widths in all months (Figure 19). SDF-derived MIZ
widths were deepest in September, and narrowest in February. Using the SDF technique, MIZ widths of in excess of 600 km
are observed to occur in May, September and December, whereas SIC-based estimates of MIZ width never exceed 200 km.
We note that SIC-derived MIZ width is wider in May than September, in contrast to the IS-2-derived result. Noting that the
Southern Ocean $H_s$ is higher in September than in May (Young et al., 2020), this may indicate that the IS-2-derived MIZ width
is a more realistic representation of waves within ice, however wave-ice interaction model studies are required to confirm this.

## 5 Discussion

### 5.1 Towards improved definition of the MIZ

MIZ width estimates from SIC were far narrower than those from IS-2, and the disparity between SIC and wave presence in sea
ice was illustrated in the case studies demonstrated in 4.1 and 4.2. It was then shown that this result occurs in all four months
studied. This work is further evidence that SIC-based MIZ may not accurately reflect the presence of waves. Both Vichi et al.





**Figure 18.** Comparison between SIC- and SDF- derived MIZ width estimates for each month. SDF MIZ width estimates were derived using a linear fit to the attenuation curve of $H_s$ estimates. Slope (m) and number of samples (n) for each month are presented. The solid black line shows linear best fit forced through the origin, and the dashed line represents the 1:1 reference. All panels range from 0 to 400 km except panel a (0 to 120 km)

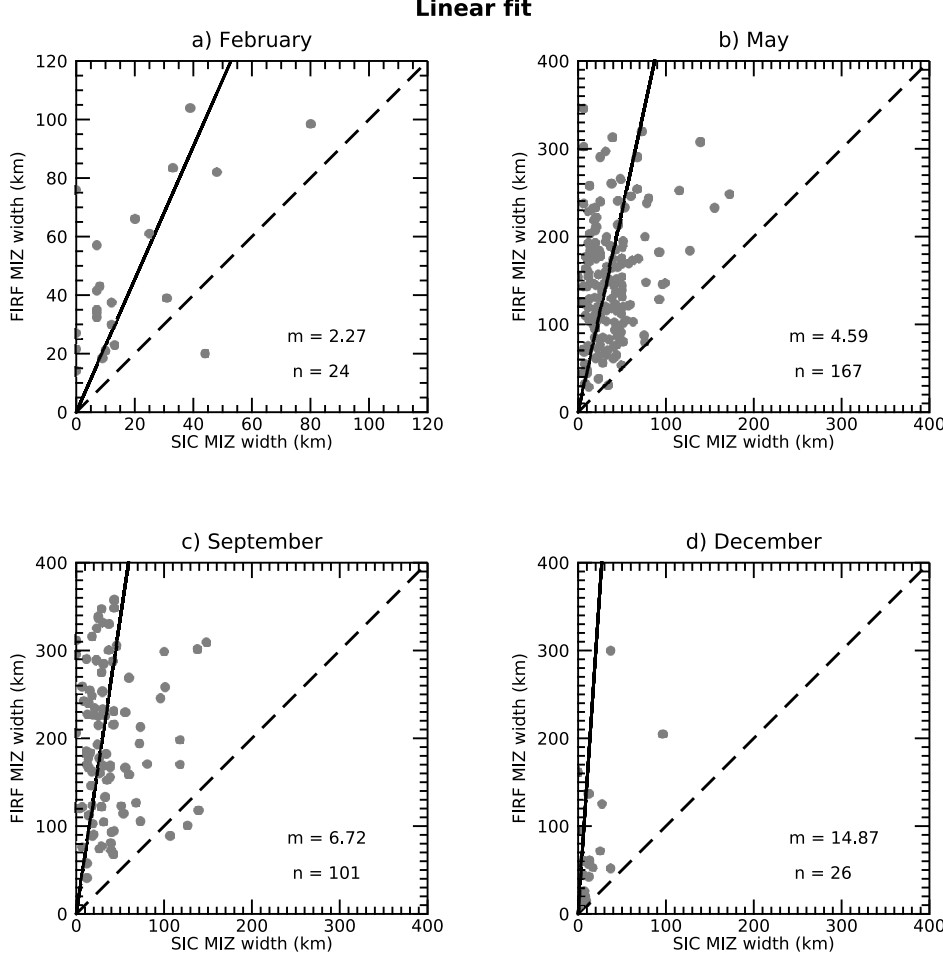





**Figure 19.** IS-2-derived MIZ width estimates by month for the the SDF methods (assuming linear and exponential attenuation) compared to the SIC-derived MIZ widths.

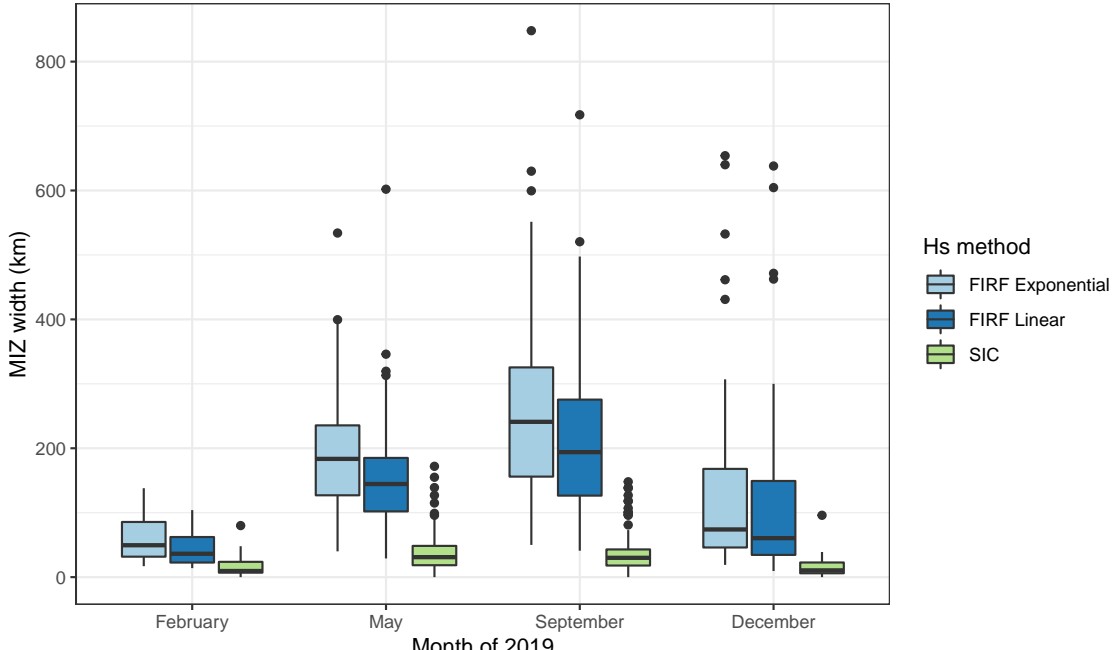

(2019) and Alberello et al. (2021) detected waves in unconsolidated yet high concentration (100%) sea ice, through which waves could easily propagate. Knowledge of SIC distribution alone (without ice type or thickness information) is inadequate
for understanding the evolution of MIZ, especially during an extreme polar cyclone (Vichi et al., 2019) or compaction events (Massom et al., 2018), where the presence of strong on-ice winds may lead to wave penetration within extremely compressed (100%) SIC. This new technique presents a potential alternative to the SIC based definition of MIZ, and may allow the study of MIZ response to extreme wave events in more detail.

In contrast to the findings presented here, Horvat et al. (2020) found IS-2 derived "wave affected " regions had a smaller
spatial extent than SIC MIZ across all seasons and hemispheres. We suggest two potential reasons for this discrepancy. First, the Horvat et al. (2020) method cannot record waves with smaller amplitude than sea ice freeboard variability, and likely underestimates the areas where waves are truly present. Second, Horvat et al. (2020) developed a gridded product from all IS-2 tracks, whereas we focused on a subset of IS2 tracks in which waves were manually identified. Thus while there is high correlation between both estimates, the inclusion of many tracks without active waves into a gridded product reduces the overall
extent of the wave-affected MIZ.

The inner MIZ boundary was here defined as the point where $H_s$ became smaller than the estimated error in the surface heights. These estimates of MIZ width (especially when assuming exponential decay) likely include small amplitude waves in the inner MIZ that may have energies too small to "significantly impact the dynamics of sea ice", as per the physical definition





of the MIZ (Weeks, 2010). It is then necessary to consider what magnitude of wave energy (or $H_s$) has sufficient impact on

sea ice properties. Sutherland and Dumont (2018) defined the MIZ as the distance from the ice edge over which modelled compressive forcing from wave stress was greater to or equal to that from wind stress. Through the techniques presented here, this "dynamics-impacted" MIZ could be studied using an adjusted (higher) $H_s$ threshold.

## 5.2  $H_s$ estimation method and caveats associated with this work

The high correlation among IS-2 MIZ distance estimation methods indicated that MIZ width estimation was relatively insensi-

tive to $H_s$ calculation method. The selection of a wavelength subset for the SDF median estimation of $H_s$ did not significantly influence MIZ estimation distances, as similar results were shown for the $H_{m0}$ estimates (i.e., integrated over 16 to 1500 m; wave periods from 3 to 30 s), and the standard deviation-derived $H_s$ (where no bandpass filtering was applied). This indicated suitability for the choice of SDF-based selection, with the potential benefit of increased sensitivity to surface gravity wavelengths most commonly present in the Antarctic MIZ and most likely to impact sea ice.

There are a number of caveats associated with the MIZ width estimation techniques presented here. Although most manual rejection of IS-2 tracks was because of cloud obscuration of the MIZ, there may be other cases excluded because no attenuation visibly occurs, such as due to quiescent wave conditions north of the ice edge. In such cases, SIC-derived MIZ estimates may be wider than those based on attenuation. This selection bias favours obvious attenuation occurring when open ocean $H_s$ is large. In these (unrepresented) low offshore $H_s$ cases, an SIC-based definition of MIZ width is also inappropriate.

A bias may also be introduced by exclusion of tracks where wave attenuation occurs from the ice edge to the Antarctic continent. Such cases are likely to occur around the time of the sea ice minimum, particularly in East Antarctica and the Bellingshausen Sea, where sea ice extent is lower than in the Weddell or Ross seas. In addition, it is possible that times of northerly winds may bring both high apparent $H_s$ at the ice edge (favouring visible and obvious attenuation) and a compacted ice edge, resulting in a narrower SIC-derived MIZ width estimate. Consideration (and elimination) of such potential biases

should form the focus of subsequent work.

## 5.3  Wave attenuation model

Exponential decay, with decay rate as a function of wave frequency, is widely accepted as an appropriate form for modelling wave attenuation (Meylan et al., 2018), is predicted by linear theory (Squire, 2020), has been demonstrated by observational wave buoy studies (Kohout et al., 2020), and has been implemented in mathematical models (Meylan et al., 2018). However,

the mean RMSE and MIZ uncertainty statistics presented here were smaller for linear than exponential fit. This may suggest non-linearity in energy transfer during attenuation, perhaps caused by variation in ice thickness, which has been shown to strongly affect attenuation rates (De Santi et al., 2018), or energy input from wind to waves in unconsolidated sea ice, which is not currently accounted for in contemporary wave models (Rogers et al., 2020). The aforementioned potential bias toward times of larger wave heights may also play a role in the apparent better fit of linear attenuation (Kohout et al., 2014; Montiel et al.,

2018), due to the occurrence of nonlinear dissipation mechanisms including wave over-wash of floes and breaking (of steep waves) close to the ice edge (Squire, 2018). Seasonal effects on attenuation may also be expected due to seasonal variability





in ice cover and type (e.g., Doble et al., 2015). The prevalence of thinner, unconsolidated pancake ice during the Antarctic growth season may result in lower attenuation rates. Meylan et al. (2014) found no change in $H_s$ within the outer 80 km of Antarctic MIZ in September, where there were small floes (10 to 25 m diameter) present and low SIC. Non-linear viscous
dissipation may be a dominant mode of wave attenuation in unconsolidated ice types (Squire, 2020). Correction for incident wave direction (Kohout et al., 2020), and the consideration of noise effects (especially at lower frequencies; Thomson et al., 2021) should also be considered in future work to ensure accuracy in attenuation coefficient retrieval.

## 5.4 MIZ seasonality

MIZ width seasonality agreed broadly with that expected from seasonal trends in Southern Ocean $H_s$, with larger incident
waves in winter months able to penetrate further into the MIZ, matching the seasonality of Young et al. (2020). We draw attention to a large discrepancy between SIC- and IS-2-derived MIZ widths: SIC MIZ width was 4.6 and 6.7 times lower than linear SDF-derived MIZ width in May and September, respectively. SIC-based studies of MIZ seasonality have shown a peak in MIZ area in October/November, after the annual sea ice area maximum (Uotila et al., 2019), whereas the wave-affected fraction metric described by Horvat et al. (2020) was unexpectedly low in November 2018. Although only four months were
assessed in the present study, improved automation of these techniques and their application over the whole IS-2 data record (September 2018 to present) may improve our understanding of seasonal MIZ dynamics, and should be prioritised for future work.

Seasonality and sea ice characteristics also play a role in the reliability of MIZ estimates. Thicker multi-year or first year ice near the ice edge in summer increases the difficulty of differentiation between wave presence from ice structure, necessitating
manual removal of such cases. By way of comparison, March to September is characterised by pancake ice present at the ice edge, providing a more homogeneous environment in which to observe attenuation. In the period of rapid spring ice retreat (November to early February), further challenges to accurate MIZ estimation occur due to the complex ice edge morphology, and the development of open water regions near the continent. Complexity in the spatial distribution of retreating ice edge can result in the orientation of IS-2 tracks not being orthogonal to the ice edge, and the measurement of waves along the boundary
of sea ice rather than into the pack, giving spurious estimates for attenuation. The variable distribution of open water at this time may increase the incidence of observing multiple wave directions due to local wave generation, potentially impacting the reliability of attenuation estimates. The highest disagreement between MIZ and SIC was observed in December, and this is likely linked to such complexities during rapid retreat. The lowest disagreement between MIZ and SIC was observed in February, likely because of the smaller total sea ice extent at this time (Eayrs et al., 2019).

## 450 5.5 Further improvements

In addition to those previously suggested, one large improvement to this work would be the consideration of incident wave direction in spectral estimates. A north-south wave propagation direction has been assumed in most previous studies on wave attenuation (e.g., Kohout et al., 2014), and was similar to the along-track wave direction assumed here. If the incident wave direction was offset from the IS-2 track, the wavelength would be underestimated by a factor of $cos(\theta)$ where $\theta$ is the angle be-



tween the incident wave direction and the satellite track. This potentially results in invalid assumptions for the SDF wavelength choice, leading to a misrepresentation of these parameters. Wave direction corrections, for example following the methods in Kohout et al. (2020), would allow physically-meaningful retrievals of wavenumber and spectral width characteristics. Wave direction will also affect the corrected distance calculation using SIC values retrieved, enabling them to be representative of SIC that waves travelled through. As there remain uncertainties in wave reanalysis products (especially within sea ice), remotely-

sensed wave direction from the forthcoming SWOT satellite radar altimeter (Armitage and Kwok, 2021) may provide improved wave direction estimates.

In addition to $H_s$, spectral width may be another important parameter to consider for MIZ width estimation. Both case studies presented here exhibit a prominent spectral peak associated with the presence of waves, although the magnitude of the peak wavelength varied due to the characteristics of the incident wave field. In contrast, areas of ice structure displayed a very

different signature of Brownian noise. Difference of spectral shape between these two regimes may be a more robust way of distinguishing the presence of MIZ (for example from isolated sections), rather than requiring a full track of attenuation. If this method were applied, an estimate of the Brownian noise threshold associated with ice structure (following the methods in Thomson et al., 2021) may provide a suitable cutoff for frequencies and power spectral amplitude over which to estimate spectral width. Consideration should also be given to the use of the lower-level ATL03 dataset from IS-2, which reports

individual geo-located photon reflection locations, and other techniques which do not require resampling of along-track data (e.g., non-uniform Fourier transforms), as these may preserve important spectral information lost in the production of the ATL07 segments.

## 6 Conclusions

This technique presents improvements over existing ways to determine wave attenuation in sea ice. Improved understanding

of wave attenuation in ice, facilitated by a larger number of widely-distributed records of attenuation from IS-2, may be applicable for incorporation into wave and sea ice models. Here, many tracks were too cloud-affected to give a complete record of wave attenuation. However, this technique may assist in developing such a record by validating other large-scale MIZ width estimates achieved from microwave-based remote sensing techniques, including scatterometers and the forthcoming generation of synthetic aperture radar altimeters. This attempt to remotely characterize MIZ width based on its true physical

definition (i.e. wave dynamics) may improve our understanding of the interactions in this zone, with widespread application for studies concerning MIZ ecology and physical processes.

*Code and data availability.* AMSR2-derived sea-ice concentration data (using the ARTIST Sea Ice algorithm) were obtained from the University of Bremen. ICESat-2 sea ice height data (ATL07 product) were obtained from the National Snow and Ice Data Center (NSIDC). The wave-ice buoy data are available from the NZ Ocean Data Network (doi:10.17632/22hpw2xn3x.1). Processed ICESat-2 data will be made

available as an Australian Antarctic Data Centre archive in compliance with FAIR Data Standards upon paper acceptance. Code to reproduce these results will be available and updated on GitHub upon paper acceptance.



## Appendix A: Missing data correction in spectral analysis

The spectral analysis of IS-2 data applied here seeks to obtain power spectra whose amplitudes are directly related to those of the underlying waves. This is made difficult by the (sporadic) absence of data due to cloud and the application of windowing functions. Both of these effects lead to a decrease in the sum contributing to the mean-square amplitude of the altimetric height segment. In accordance with Parseval's identity, the average spectral power also decreases according to

$$\frac{1}{N} \sum_{k=0}^{N-1} |c_k|^2 = \sum_{n=0}^{N-1} |C_n|^2, \tag{A1}$$

where $c_k$ are the mean-removed heights (potentially attenuated or set to zero) along a segment of the track containing $N$ points and $C_n$ are the components of its digital Fourier transform (DFT). (Note that the position of the $1/N$ factor depends on the form of the DFT being used.)

Following Press et al. (1992, section 13.4), compensation for the effect of windowing can be achieved by replacing the $1/N^2$ factor used to calculate spectral amplitudes from $|C_n|^2$ with $1/W_{ss}$, where

$$W_{ss} \equiv N \sum_{j=0}^{N-1} w_j^2, \tag{A2}$$

and $w_j$ is the window function weighting applied to each $c_k$ before the application of the DFT.

The usual application of a windowing correction sees window amplitudes $w_j$ summed over all $N$. In the presence of missing data, the windowing function can be reinterpreted to be the product of the windowing function and a missing data mask, equal to unity only where valid data are present. To adapt the $W_{ss}$ correction, an interpretation where $w_j$ is zero within data gaps is applied, such that

$$W_{ss} \equiv N \sum_{j \in G} w_j^2, \tag{A3}$$

where $G$ is the set of indices of good data. This acts to scale the spectral coefficients back up to the (physical) wave amplitudes, and to do it in a way that includes the impact of the distribution of the missing data on the windowing function.

*Author contributions.* All authors edited the manuscript. JB and ADF led the analysis, produced all figures, drafted the paper and coordinated co-authors. DJM advised on spectral and spatial analyses. PW, GDW and AK deployed the buoys and contributed to the analysis. AA and RAM assisted with wave-ice interaction considerations. CH produced the along-track wave-affected fraction estimates and contributed to the interpretation of results. SW and JC assisted with data preparation.

*Competing interests.* The authors declare no competing interests.



*Acknowledgements.* This project received grant funding from the Australian Government as part of the Antarctic Science Collaboration Initiative program, and contributes to Project 6 of the Australian Antarctic Program Partnership (Project ID ASCI000002). This research was supported by use of the Nectar Research Cloud and by the Tasmanian Partnership for Advanced Computing. The Nectar Research Cloud is

a collaborative Australian research platform supported by the NCRIS-funded Australian Research Data Commons (ARDC). We thank the NSIDC and University of Bremen for providing the ICESat-2 and sea ice concentration data, respectively. We are deeply grateful to the captain, officers, crews, and scientists on board icebreaker Shirase for their help with field observations for the 61st Japanese Antarctic Research Expedition (JARE). AA acknowledges support from the Japan Society for the Promotion of Science (PE19055). PW also acknowledges support from the Japan Society for the Promotion of Science (18F18794).





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
