# Peer review of "Altimetric observation of wave attenuation through the Antarctic marginal ice zone using ICESat-2"

_The Cryosphere, 2021_

## Referee Comment (RC2)

This paper documents the process of observing ocean surface gravity waves in sea ice and the estimation of MIZ width based upon the wave height measurements. The initial observations used are surface elevation measurements using the laser instrument on ICESat2. Individual elevation measurements within sea ice are combined through several spectral analysis techniques in order calculate the wave heights within sea ice. A clear difference in elevation spectrum is presented between the MIZ and thicker ice further from the sea ice edge. There is a clear region of decreasing significant wave height within the sea ice pack for the case studies presented. The attenuation rate of the significant wave height is successfully modelled an estimate of MIZ width is presented. The satellite based observations are compared to in situ buoy measurements of wave height and the implications of wave based measurements of are discussed.

I find that the paper is well written and the results are carefully discussed and contrasted to existing research. I enjoyed reading the discussion and conclusions sections. These new results showing observed wave attenuation rates within sea ice are of high importance to current research and I recommend the paper for publication after the authors deal with some editing concerns.

First I found that my reading of the paper was greatly hampered by the volume of data presented in the paper. In particular I recommend that the authors remove many of the figures to supplemental material. Each case study needs only 2 figures within the paper body. One for the initial data, and another for the retrieved parameters, for example. Currently each study has 7 figures. The inclusion of 7 different analysis techniques and 3+ plots for each technique for both case studies, and then a further figure inter-comparing these techniques these techniques (figure 17) when they appear to produce very similar results, makes the the reading of the paper a chore and lessens the impact of what is a high quality paper. Further in the paper a single FIRF wave length analysis was presented and the best to use for MIZ width estimates. Please consider presenting the case studies for this method only, with a table or figure summarising the other methods and the extended figures in supplemental material.

My second concern is with the method of MIZ width estimation. Currently the authors use the point at which the modelled wave height drops below the model error. However, there is no data that is apparent at this point, as the authors point out that other effects then dominate the signal, such as ice dynamics. As the authors state that the extent of the MIZ is the region where wave energy is of greater importance than ice dynamics, then surely the 'first local minima' in figure 5, or the 'break point' in figure 6, are also valid points at estimating the MIZ width. The measurements for the case studies show will also give a MIZ the exceeds that from Ice concentration. Including them as other estimates of MIZ width in the rest of the study (say in figure 18 and 19) will improve the strength of the results.

Specific points:

L 5 Are these new techniques presented within this paper?

L31 are you saying that long-period wave impact the size distribution far into the sea ice pack? This sentence is long and hard to follow.
L33 SO int he southern ocean wave do not penetrate? These tow sentences seem to contradict each other
L80 New paragraph here will enable readers to find the purpose description more easily.

L 95, an extended description of the paper contents here will be useful This is a long technical paper and a description of every section and key subsections will ease the navigation of it.

L129 what is a 'segment height'?

Figure 4 Caption 'distance to the continent' is 'distance from the ice edge' surely?

Figure 2. What benefit is there in having all three FIRF wavelengths presented here and in further figures? Is it possible to pick a best use case, and allow further description in supplemental material?

Figure 2. Please put the acquisition time and date in every caption where there is a figure representing data from a single track. This makes it much easier to cross reference to other studies and to make sure the reader is comparing like for like about the paper.

L275, I'm struggling to see how the peak shift towards longer wavelengths further from the ice edge. Figure 4, top pane, has a peak for all colours at around wavenumber = 0.001. Is this what you mean? I can see no significant movement in this peak.

L277, there is no clear peak the second pane in Figure 4,

L 281. I am struggling to see how figures 5-7 describe the model. There are three figures here with lots of detail that is not referred to in the paper. For example the letters a-g for figure 5 are not mentioned. Is it possible that these three figures can be abbreviated into one for the paper with the full figures moved to supplemental material?

Figures 6-7 I cannot find a description for how the red line is generated. Is it the point of intercept for the line of best fit in Hs? Please refer to this in the description in section 4.1.

Figure 6, you do not refer directly to what exact value is shown by the grey scatter points. Are they identical to the grey points in figure 5?

Figure 8. This figure without the previous figures 5-7 may be all that is needed for the main paper.

Line 305. I don't not find it helpful for 19 lines of text to be supported by 7 figures. It makes the paper very cumbersome to read, and lessens the impact of the results. Please re edit the figures to have two figure per track. Perhaps illustrating a single processing option, Hann Hm0, or a single FIRF, with the rest in supplemental material.

L 320, is it possible to give the distance from each buoy to the ice edge?

L 322, you present 7 techniques for obtaining Hs from IS2, which was used for this comparison?

L 328, for the satellite to buoy distance measurement, are you able to comment on whether the satellite measurements are closer to the ice edge than the buoy, or vice versa? Knowledge of this is helpful for estimating whether the satellite is underestimating wave height, or whether the buoy is in a location of higher waves. The distances referred to in this section are larger than the MIZ extents presented elsewhere in the paper. Is it possible to use ice concentration data to get a distance from the buoy to 15% concentration? Also it would be nice to see the time series of buoy records against the coincident satellite retrievals, is this possible to display?

L328, why does the regression slop indicate that there is an underestimation? Please expand. My interpretation of this is that there is less of a correlation in measurement as separation increases, which is difficult to interpret if the distance from the buoy to ice edge is not considered.

L336 if a single technique is chosen, then please present only that one in the main paper. The rest is only helpful in supplemental material as there are far too many repeated plots.

L340 This suggest that only one SDF technique is required. Please reduce the main paper to include only that one.

Figure 11, again, does the caption mean to say 'distance from the ice edge' instead?

L 344, I'm not surprised that all these techniques agree, as it is very hard to distinguish between all the previously displayed data.

Figure 13, the MIZ width estimates for linear model appear to be the linear intercept, what is the definition for the log scale model? I am struggling to find the definitions in the paper body.

Figure 19. This caption needs expanding. What are the black dots? Why are they not included in the February IS2 cases? What statistical values are used to create the boxes?

L 380, In all the examples you show, your "wave affected" region exceeds the SIC based MIZ in all cases, what are the reasons, physical and technical, for the Horvat et al. region having a far smaller spatial extent?

L 386 Ok this is your definition of MIZ width. This definition was not clearly defined early in the paper and it is frustrating to find it so late in the paper. What is the 'estimated error'? Is this calculated per track, or a constant parameter? The error displayed in the plots is highly variable, particularly in the log models.

---

## Author Response (AR1)

Our responses in **bold**.

Reviewer 1: Dr Fabien Montiel

The manuscript describes a new method to estimate the attenuation of ocean waves in the ice-covered Southern Ocean using altimetric data from ICESat-2. The main application is to estimate the width of the marginal ice zone (MIZ), as the spatial extent over which wave attenuation is observed. Given the computationally heavy process involved, only four months of data is analysed. The authors strongly emphasize the advantage of their method compared to another widely-used definition in terms of sea ice concentration.

In my opinion, this is a very good paper that deserves to be published in Cryosphere. It is well written, the work is novel and very rigorously presented, and the results/discussion are interesting. Despite my lack of expertise in remote sensing technology and data analysis, I managed follow most of the methods section. Although my recommendation is for publication with minor revisions, I would like the authors to address the following comments.

**We thank Dr Montiel for recognising the importance of the study, and for his thorough review of the manuscript.**

Main comments

1. My first and basically only concern relates to the need to consider "MIZ width" as a precise quantity in the first place. I understand there has been some previous work on trying to somehow measure precisely the MIZ and find its "boundary" with pack ice. I am skeptical about this as the MIZ was never well defined. The authors quote the "definition" of the MIZ from Wadhams in the introduction (lines 24-25), which is clearly qualitative at best. Any attempt to quantify it will therefore be up the authors to come up with a metric, be it concentration-based or wave attenuation-based. I don't see any reasons why we should expect these would match as they measure different things. In my opinion, the danger in this exercise is to characterise the sea ice cover in a binary manner, i.e. MIZ or not MIZ. I feel like what we are really after is more nuanced, again especially referring to the non-quantitative definition of the MIZ. I fully agree with the authors that the concentration-based definition is lacking as it does not consider "open-ocean processes". At the same time, a wave attenuation-based definition also has some issues. For instance, if there are temporarily no waves, does the MIZ stop existing during that time? I want to be clear that I am not criticising the work of the authors in trying to quantify the spatial extent over which wave attenuation is observed. This is very interesting and the method they use clearly has a lot of potential for other applications. My concern is more trying to qualify this metric as the definition for the MIZ width. So when the authors refer to "the true MIZ extent" (line 58), the "physical definition of the MIZ" (lines 388-389) or "its true physical definition" (lines 479-480), I am arguing that this is an ill-defined concept and that there is no such thing as a true definition of the MIZ. If there was one, it surely would depend on ice properties as well as wave characteristics. My suggestion for the authors is therefore to rephrase some parts of the manuscript so

as to incorporate the fact that MIZ and MIZ width are qualitative concepts as opposed to well defined quantities, unless of course they have a counter argument which I would be very interesting in reading.

**We completely agree that there are a number of subtleties related to the MIZ definition, and that there are ways that we could improve how this is communicated in our manuscript. Indeed, what we have done is demonstrated that ICESat-2 can retrieve the limit of wave penetration at a snapshot in time. We agree that this may or may not truly represent some binary measure of the MIZ, for a number of reasons, including a) heavily attenuated wave passage may not alter the physical properties of the sea ice, even if it's still observable with ICESat-2; b) even with perfect observation of the penetration of waves, the wave penetration "yesterday" (or in the recent past) may have been higher, so that ice may still be "modified by interaction with the ocean" (i.e., MIZ). We do concede that what we have measured here may not be the true MIZ extent for these reasons, and that a binary view of MIZ may be unattainable (and may even differ for different purposes). We thank Dr Montiel for bringing up this important point and have both a) softened language around "retrieving the MIZ" throughout the manuscript (largely replaced with "wave penetration width"), and b) added a section in the introduction that more clearly defines this and "SIC-derived MIZ width" estimation techniques in more detail, and acknowledges the qualitative nature of defining this region. We have also edited the quotations you have provided and the discussion section to reflect this.**

2. The authors seem not to have considered the modelling work of Tim Williams, Danny Dumont and Luke Bennetts on MIZ width as measured by the extent of the ice cover over which wave-induced breakup can occur (see, e.g., Dumont et al, 2011, JGR; Williams et al, 2013a,b, Ocean Model.; Bennetts et al., 2014, Ann. Glaciol., Williams et al., 2017, Cryosphere). I feel this work needs to be discussed as they used another wave-based criterion to measure the extent of the MIZ and is therefore more in line with the proposed definition than the SIC-based one. Of course my previous comment still applies to this other definition of MIZ width.

**Thank you for pointing out this deficiency in the literature covered - we fully recognise the potential for scientific advance by combining techniques such as ours with modeling wave attenuation studies (and indeed, have plans to contribute to work being planned by Luke Bennetts in this area). We have added a sentence in the introduction and also in the discussion section 5.1.**

Other comments/typo

3. line 21: r missing in "anthropogenic".

**Fixed**

4. line 33: the authors might want to consider including Montiel et al. (2022), which has analysed the largest dataset to date of in situ wave buoy measurements in the SO, as another reference. The paper has just been accepted in JPO and can be accessed on arXiv at https://arxiv.org/abs/2111.04819.

**We have incorporate it into the literature review in the revised manuscript (alongside the Squire 2020 reference). Such a dataset will be of considerable importance to wider validation studies.**

5. line 84: This sentence is circular as it essentially says that estimating MIZ width improves knowledge of MIZ width!

**The sentence has now been amended to remove circulatory.**

6. Eq. (1) and line 150: I am a bit confused by this metric and why the authors use it to measure attenuation. Could the authors please clarify?

**This metric is an attempt to measure the total amount of ice along any particular transect by "compacting" the lower concentration ice to 100% ice, then measuring the width of that transect of 100% ice. It is used here because wave attenuation is very low in regions of open water, so the important metric is the total length of 100% ice equivalent. We have now clarified this in the revised manuscript - thanks for pointing out that it wasn't clear.**

7. line 191+: I believe "change-point" is more appropriate than "breakpoint".

**The "breakpoint" terminology comes from the "segmented" package in R. We now note, after checking this, that the term "change point" seems to occur interchangeably in their documentation. Given the reviewer's strong Mathematics background, we are very happy to defer to their expertise, and have changed it as suggested in the revision.**

8. line 212: "on" missing.

**Has been added**

9. line 325: I don't think n has been defined previously.

**Very good point - has now been amended.**

10. line 363: remove "of".

**Has been removed.**

11. line 366-367: I'm not sure I understand the statement "this may indicate ... within ice". Since you defined the MIZ based on waves, is it not necessary that waves are present in the MIZ?

**Reading this back, I agree that it's very ambiguous. We tried to argue that the seasonality of the wave climate aligned broadly with ours, so this may suggest that our retrievals are accurate. This sentence has been rectified in the revised manuscript.**

12. line 474: that is a bold statement. Not sure what it is based on as there have not been any comparisons with other approaches to measure wave attenuation done in this paper. Consider removing or better justifying this statement.

**To be clear in this response, I believe you are referring to the sentence "This technique presents improvements over existing ways to determine wave attenuation in sea ice." Reading this back, I can see how it can be interpreted in unintended ways. We believe that this technique is currently the most precise way of currently remotely retrieving wave attenuation in sea ice. As it currently stands, the sentence (unintentionally) encapsulates in situ techniques as well. We have revised this in order to emphasise our intention.**

==============================================================

Our responses in **bold**.

Reviewer 2: Dr Harry Heorton

This paper documents the process of observing ocean surface gravity waves in sea ice and the estimation of MIZ width based upon the wave height measurements. The initial observations used are surface elevation measurements using the laser instrument on ICESat2. Individual elevation measurements within sea ice are combined through several spectral analysis techniques in order calculate the wave heights within sea ice. A clear difference in elevation spectrum is presented between the MIZ and thicker ice further from the sea ice edge. There is a clear region of decreasing significant wave height within the sea ice pack for the case studies presented. The attenuation rate of the significant wave height is successfully modelled an estimate of MIZ width is presented. The satellite based observations are compared to in situ buoy measurements of wave height and the implications of wave based measurements of are discussed.
I find that the paper is well written and the results are carefully discussed and contrasted to existing research. I enjoyed reading the discussion and conclusions sections. These new results showing observed wave attenuation rates within sea ice are of high importance to current research and I recommend the paper for publication after the authors deal with some editing Concerns.

**We thank Dr Heorton for his very thorough consideration of the manuscript, including encouraging remarks as well as both scientific and structural improvement suggestions.**

First I found that my reading of the paper was greatly hampered by the volume of data presented in the paper. In particular I recommend that the authors remove many of the figures to supplemental material. Each case study needs only 2 figures within the paper body. One for the initial data, and another for the retrieved parameters, for example. Currently each study has 7 figures. The inclusion of 7 different analysis techniques and 3+ plots for each technique for both case studies, and then a further figure inter-comparing these techniques these techniques (figure 17) when they appear to produce very similar results, makes the the reading of the paper a chore and lessens the impact of what is a high quality paper.
Further in the paper a single FIRF wavelength analysis was presented and the best to use for MIZ width estimates. Please consider presenting the case studies for this method only, with a table or figure summarising the other methods and the extended figures in supplemental material.

**Your point highlights a major consideration/discussion point among the authors during the writing of the manuscript: to use a supplement or to avoid it? In fact, your solution represents "the other choice": key figures in the main text but figures of lesser importance moved to a supplement (a similar permutation was also discussed internally before submission!)**

**In the end, we decided to retain all figures in a "linear" style narrative. Although this structure didn't draw criticism from the other reviewer, we can clearly see the drawback of this choice, as you clearly outlined.**
**I can't fault your suggested structure too much (there was considerable internal debate about this!) - it retains key information in the main text while keeping the details available for those who want a deeper dive. We have incorporated these suggestions in a revised structure for the revised manuscript. In particular: Now only two figures remain in the main text for each case study (key figures remaining are the central beam heights and the spatial domain filtered time series of this; and the figure comparing central beam heights with along-track SIC and the Hs derived from spectral techniques). The other figures are now in appendices. Thanks for taking the time to suggest this structure.**

My second concern is with the method of MIZ width estimation. Currently the authors use the point at which the modelled wave height drops below the model error. However, there is no data that is apparent at this point, as the authors point out that other effects then dominate the signal, such as ice dynamics. As the authors state that the extent of the MIZ is the region where wave energy is of greater importance than ice dynamics, then surely the 'first local minima' in figure 5, or the 'break point' in figure 6, are also valid points at estimating the MIZ width. The measurements for the case studies show will also give a MIZ the exceeds that from Ice concentration. Including them as other estimates of MIZ width in the rest of the study (say in figure 18 and 19) will improve the strength of the results.

**We are not quite sure what "there is no data that is apparent at this point" means. We explicitly (manually) excluded all tracks where the transition region was obscured by cloud - so all tracks presented here had data in the region of transition from wave-dominated to structure-dominated Hs contributions.**
**Regarding your points (separated out here) of 1) "surely the 'first local minima' in figure 5 … [is a] valid point at estimating the MIZ width"; and 2) "[surely the] … 'break point' in figure 6 [is a] valid point at estimating the MIZ width": we did in fact consider both of these choices early in our work, but these were both ruled out. We would like to respond to these separately and explicitly:**

1) **This would indeed be a simpler method than what we chose (requiring only a GAM smooth, thus dispensing with the complicated breakpoint calculation). The problem with using the first local minimum as the MIZ width is that this is just a simple smooth of the underlying height data. Smoothing kernels have a finite width, and we could imagine circumstances in which information is needlessly lost by this simpler technique. Further, the use of the breakpoint method allows us to model the Hs attenuation with linear or exponential curves, in line with current understanding and modelling of wave attenuation in ice, as opposed to a more arbitrary smoothing curve.**

2) **The disadvantage of simply using the breakpoint is that a) the "inner" regime (ice structure-dominated part) contributes to the location of the breakpoint. This is undesirable (i.e., ice ~2*breakpoints from the edge can influence the location of**

the breakpoint, but is far from the limit of wave penetration so should not influence the location of the boundary). Our method, while more complex, avoids this problem.**

**We have updated the "Attenuation curve fitting and wave penetration width estimation" part of the methods to clarify these points. We have also updated the case study appendices to describe this process with references to the relevant figures.**

Specific points:
L 5 Are these new techniques presented within this paper?
**Yes - the GAM smoothing -> breakpoint estimation -> combination is new, even if the FIRF technique has been applied to other problems elsewhere before.**

L31 are you saying that long-period wave impact the size distribution far into the sea ice pack? This sentence is long and hard to follow.
**Yes, this was our intent, and this has been shown in Kohout et al 2014, for example, although this work was not referenced in this sentence (we plan to add it). To be clear in this response, we assume you are referring to the sentence beginning with "In some cases…" (i.e., the sentence ending on L31). We agree that it could be clarified and shortened, and have done so in the revision.**

L33 SO int he southern ocean wave do not penetrate? These tow sentences seem to contradict each other
**Oh, our intent with that "attenuate" was something like "penetrate and attenuate" - rather than "become attenuated before significant penetration". Apologies for the ambiguity - this has been resolved in a revised submission.**

L80 New paragraph here will enable readers to find the purpose description more easily.
**Agree - has been added.**

L 95, an extended description of the paper contents here will be useful This is a long technical paper and a description of every section and key subsections will ease the navigation of it.
**This is a good point. Normally I avoid adding a structure sentence unnecessarily, but I agree that this manuscript would benefit from it. It has been added in the revision.**

L129 what is a 'segment height'?
**This is the terminology from the ATL07 product. A segment is a collection of returned photons over a finite distance (the distance varies from several metres to several tens of metres). Thus, segment height is the height of this segment, and is the fundamental quantity reported in ATL07. This information has been added to this sentence.**

Figure 4 Caption 'distance to the continent' is 'distance from the ice edge' surely?
**Yes - thanks for picking this up. Has been rectified.**

Figure 2. What benefit is there in having all three FIRF wavelengths presented here and in further figures? Is it possible to pick a best use case, and allow further description in supplemental material?

**There are four FIRFs presented here. We use the median of these four FIRFs when we use the SDF technique. The rationale behind using four FIRFs is given at around line 335. This has been clarified about L340.**

Figure 2. Please put the acquisition time and date in every caption where there is a figure representing data from a single track. This makes it much easier to cross reference to other studies and to make sure the reader is comparing like for like about the paper.

**A good suggestion - has been added.**

L275, I'm struggling to see how the peak shift towards longer wavelengths further from the ice edge. Figure 4, top pane, has a peak for all colours at around wavenumber = 0.001. Is this what you mean? I can see no significant movement in this peak.

**An earlier version of this figure had a linear x-axis, and the peak shift was far more readily apparent without the log transformation. While we still believe this shift is a robust feature, we concede that it's no longer apparent in the current figure version. This peak shift wording has been removed from the revision.**

L277, there is no clear peak the second pane in Figure 4,

**This is correct - we don't expect a clear wave-related peak since waves have been fully attenuated here. As above, the text at line 274 has been amended to remove the wording around peak shifts, which will remove the ambiguity here too.**

L 281. I am struggling to see how figures 5-7 describe the model. There are three figures here with lots of detail that is not referred to in the paper. For example the letters a-g for figure 5 are not mentioned. Is it possible that these three figures can be abbreviated into one for the paper with the full figures moved to supplemental material?

**You're right in saying that many of the figures here have scant explanation. Part of the difficulty is that we're using the median of four spatial domain filters, so these probably need to be present, even if each one is not discussed in detail. While we're including the four SDF diagrams (for Figure 5), it seems that we might as well include the other techniques discussed. We like your idea to only include SDF/FIRF figures in the main text, and move the others to supplementary material. This has been done as part of the restructure.**

Figures 6-7 I cannot find a description for how the red line is generated. Is it the point of intercept for the line of best fit in Hs? Please refer to this in the description in section 4.1.

**Yes this is the point of intercept. This has been added to the description (N.B. case studies are now in an appendix).**

Figure 6, you do not refer directly to what exact value is shown by the grey scatter points. Are they identical to the grey points in figure 5?

**Thanks for pointing this out. These are Hs estimates as determined by the various techniques. Yes these are the same points as Fig 5. This has been added to the figure 6 caption.**

Figure 8. This figure without the previous figures 5-7 may be all that is needed for the main paper.
**We appreciate the simplicity of this suggestion. We have implemented the separate supplement idea and cut down on the figures presented in the main text, as suggested in your major comment above.**

Line 305. I don't not find it helpful for 19 lines of text to be supported by 7 figures. It makes the paper very cumbersome to read, and lessens the impact of the results. Please re edit the figures to have two figure per track. Perhaps illustrating a single processing option, Hann Hm0, or a single FIRF, with the rest in supplemental material.
**The second case presented will naturally have a shorter explanation. But I agree that this will be rectified by implementing your structure suggestion, which we have done. The FIRF approach is now presented in the main text, with the others in the supplementary material.**

L 320, is it possible to give the distance from each buoy to the ice edge?
**Yes - this has been added.**

L 322, you present 7 techniques for obtaining Hs from IS2, which was used for this comparison?
**Good point - this information should be clearly stated. In this case it was the Hann-filtered Hm0 technique but the Hann, boxcar and std-dev techniques all had very similar correlation coefficients. We didn't consider the SDF technique for the validation part of this work since it's not straightforward to get a single Hs number from a number of SDF-filtered time series that have overlapping frequency ranges. This information has been added.**

L 328, for the satellite to buoy distance measurement, are you able to comment on whether the satellite measurements are closer to the ice edge than the buoy, or vice versa? Knowledge of this is helpful for estimating whether the satellite is underestimating wave height, or whether the buoy is in a location of higher waves. The distances referred to in this section are larger than the MIZ extents presented elsewhere in the paper. Is it possible to use ice concentration data to get a distance from the buoy to 15% concentration? Also it would be nice to see the time series of buoy records against the coincident satellite retrievals, is this possible to display?
**Yes. Since ICESat-2 has an orbital inclination of very close to 90 degrees, the alignment of the tracks is close to north-south, even with the relatively high latitude of this validation area. This means that, cloud coverage notwithstanding, the distance of nearest observation is almost always close in latitude, with most of the difference occurring in longitude. So the satellite closest satellite measurements are generally at a similar distance (from the ice edge) as the buoy (assuming that the ice edge occurs at a roughly constant latitude). We think that the anisotropy of buoy vs satellite measurement**

**decorrelation should be the focus of a (potentially very interesting) follow-up study, rather than fully fleshed out here. That the distances are larger here (x-axis up to 500 km) than elsewhere in the paper is a really good point - we do need to at least describe the geometry of the situation a little better in this section - thank you for the good suggestion. Unfortunately there are insufficient buoy-ICESat-2 conjunctions to plot this in a time series with this buoy deployment. We think it's good fodder for a follow-up study (probably requiring a longer/larger buoy deployment covering a larger area of the MIZ).**

L328, why does the regression slop indicate that there is an underestimation? Please expand. My interpretation of this is that there is less of a correlation in measurement as separation increases, which is difficult to interpret if the distance from the buoy to ice edge is not considered

**A correct estimation (i.e., not an under- or over-estimation) would have a slope (red line) of ~1, not the value of up to ~0.5 shown here - hence we have an underestimate of Hs when using the ICESat-2 techniques (compared to Hs values retrieved from the buoys). This will be made clearer in the manuscript. We agree that distance to the ice edge would have been an important parameter to consider if the satellite tracks weren't aligned mainly north-south (i.e., perpendicular to the ice edge), as described in the previous comment. Regardless, we think a more comprehensive study (with far more wave ice buoy data) is needed to really expand on this small validation study.**

L336 if a single technique is chosen, then please present only that one in the main paper. The rest is only helpful in supplemental material as there are far too many repeated plots.
**Point taken - SDF/FIRF techniques are now presented in the main text while the others will be relegated to appendices in the revised submission.**

L340 This suggest that only one SDF technique is required. Please reduce the main paper to include only that one.
**Here by "appropriate SDF selection" we mean "the median of four SDFs" - so yes, all four are needed. We have, however, moved the non-SDF techniques to supplementary material, as described above.**

Figure 11, again, does the caption mean to say 'distance from the ice edge' instead?
**Yes - thanks again for picking up on this. Has been rectified in the resubmission**

L 344, I'm not surprised that all these techniques agree, as it is very hard to distinguish between all the previously displayed data.
**We're also happy that these techniques agree so well despite their fundamental differences!**

Figure 13, the MIZ width estimates for linear model appear to be the linear intercept, what is the definition for the log scale model? I am struggling to find the definitions in the paper body.

**We simply log-transformed the y-axis and fitted a linear model to fit an exponential model. This was done since the "segmented" package, used for the breakpoint technique, can only fit straight lines. Upon searching, I now realise that we didn't include this in the methods, so it has been added to the Methods section: "attenuation curve fitting and wave penetration width estimation".**

Figure 19. This caption needs expanding. What are the black dots? Why are they not included in the February IS2 cases? What statistical values are used to create the boxes?
**Apologies - this should have been included in the caption. From the GGplot boxplot documentation: "The lower and upper hinges correspond to the first and third quartiles (the 25th and 75th percentiles). The upper whisker extends from the hinge to the largest value no further than 1.5 * IQR from the hinge (where IQR is the inter-quartile range, or distance between the first and third quartiles). The lower whisker extends from the hinge to the smallest value at most 1.5 * IQR of the hinge. Data beyond the end of the whiskers are called "outlying" points and are plotted individually." This information has been added.**

L 380, In all the examples you show, your "wave affected" region exceeds the SIC based MIZ in all cases, what are the reasons, physical and technical, for the Horvat et al. region having a far smaller spatial extent?
**These are already detailed at Line 379, beginning "In contrast to the findings presented here, Horvat et al. (2020) found IS-2 derived "wave affected" regions had a smaller spatial extent than SIC MIZ across all seasons and hemispheres. We suggest two potential reasons for this discrepancy…"**

L 386 Ok this is your definition of MIZ width. This definition was not clearly defined early in the paper and it is frustrating to find it so late in the paper. What is the 'estimated error'? Is this calculated per track, or a constant parameter? The error displayed in the plots is highly variable, particularly in the log models
**Our inner MIZ boundary is also defined at line 197 (where the estimated error is also defined) - but we agree that a definition is needed in the introduction too. This is in line with Reviewer 1's overarching comment (that a binarised definition of MIZ is probably unattainable, and the definition subtleties need to be acknowledged). The estimated error is calculated per 6.25 km section. This detail on $H_s$ error calculation is provided in the methods section: "Derivation of significant wave height". Definition of MIZ width has been added in the purpose section of the introduction.**